# EgoSchema: A Diagnostic Benchmark for Very Long-form Video Language Understanding

**Karttikeya Mangalam**          **Raiymbek Akshkulakov**

**Jitendra Malik**

UC Berkeley

{mangalam, raiymbek, malik}@eecs.berkeley.edu

egoschema.github.io

## Abstract

We introduce EgoSchema, a very long-form video question-answering dataset, and benchmark to evaluate long video understanding capabilities of modern vision and language systems. Derived from Ego4D, EgoSchema consists of over 5000 human curated multiple choice question answer pairs, spanning over 250 hours of real video data, covering a very broad range of natural human activity and behavior. For each question, EgoSchema requires the correct answer to be selected between five given options based on a three-minute-long video clip. While some prior works have proposed video datasets with long clip lengths, we posit that merely the length of the video clip does not truly capture the temporal difficulty of the video task that is being considered. To remedy this, we introduce temporal certificate sets, a general notion for capturing the intrinsic temporal understanding length associated with a broad range of video understanding tasks & datasets. Based on this metric, we find EgoSchema to have intrinsic temporal lengths over $5.7\times$ longer than the second closest dataset and $10\times$ to $100\times$ longer than any other video understanding dataset. Further, our evaluation of several current state-of-the-art video and language models shows them to be severely lacking in long-term video understanding capabilities. Even models with several billions of parameters achieve QA accuracy less than 33% (random is 20%) on the EgoSchema multi-choice question answering task, while humans achieve about 76% accuracy. We posit that EgoSchema, with its long intrinsic temporal structures and diverse complexity, would serve as a valuable evaluation probe for developing effective long-term video understanding systems in the future. Data and Zero-shot model evaluation code are open-sourced under the Ego4D license at egoschema.github.io.

## 1 Introduction

We introduce EgoSchema, a diagnostic benchmark for assessing very long-form video-language understanding capabilities of modern multimodal systems. Understanding long natural videos requires a host of interconnected abilities such as action and scene understanding, perceiving and tracking object states, long-term visual memory, abstract reasoning, hierarchical information aggregation, and more. Shown in Fig. 1 is an exemplar of the curated EgoSchema dataset. Consider the visual cognitive faculties involved in answering the question: 'What is the overarching behavior of C and the man in the video?'. First, is the spatial recognition capabilities for disambiguating the

37th Conference on Neural Information Processing Systems (NeurIPS 2023) Track on Datasets and Benchmarks.

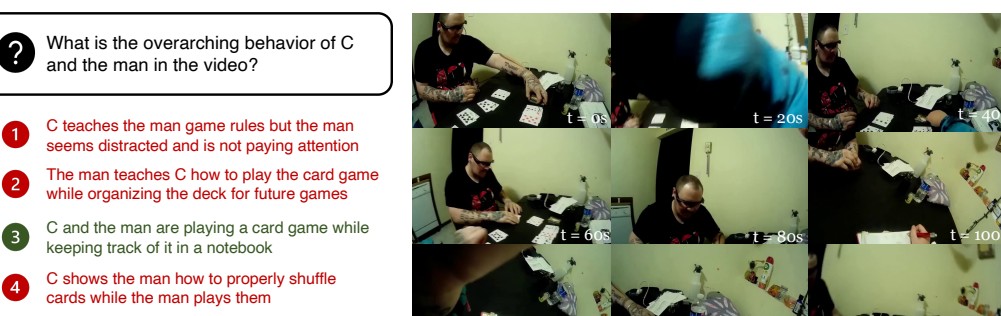

Full Video Link: youtu.be/DIyyVccQPbg          Full Video Link: youtu.be/Tp4q5GeHVMY

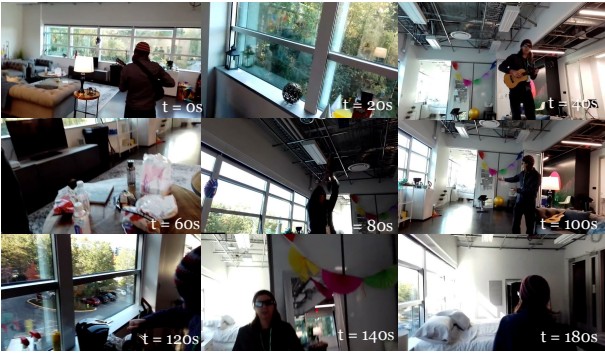

Figure 1: **The EgoSchema dataset** contains over 5000 very long-form video language understanding questions spanning over 250 hours of real, diverse, and high-quality egocentric video data. Each question requires choosing the correct answer out of five choices based on a *three minute* long video clip. The questions are manually curated to require very long *temporal certificates* (§3.2). EgoSchema median certificate length is about 100 seconds, which is 5× longer than the closest second dataset and 10× to 100× longer (Fig. 3) than any other video understanding dataset. State-of-the-Art video-language models consisting of billion of parameters achieve very low accuracy (< 33%) in Zero-shot evaluation (random is 20%) while humans achieve about 76%. 'C' refers to the camera wearer. Visualized clips are available at egoschema.github.io/explorer.

referred character 'C' (camera wearer) and 'the man' as well as the present objects such as 'cards', 'notebook', deck as so on. Next is short-term temporal recognition capabilities of understanding the atomic actions and movement of the characters such as 'playing', 'taking notes', 'shuffling' etc. Built upon these are the capabilities for visually understanding the mental states such 'distracted', 'attention' and social dynamics such as 'teaching', 'showing'. Next are medium-term actions such as 'organizing the deck' or 'keeping track'. Finally, long-term reasoning capabilities need to be employed for abstracting the 'overarching behavior' of the video from all the low-level signals to be able to rule out all the other wrong options and conclude option 3 to be correct. Note that even for humans, it is impossible to answer the illustrated questions with only the shown 9 uniformly sampled frames from the three-minute video (Fig. 1).

While there have been some prior attempts to formulate long-form video tasks [51, 44], they broadly tend to fall into two failure modes. The first failure mode stems from the difficulty of capturing the explosive diversity of human behavior in narrow pre-defined label spaces that leading unduly narrow and oddly specific tasks, such as like ratio or relationship prediction [51]. Hence, we propose to probe video systems capturing the rich complexity of long-form video with something just as rich and complex – natural language. However, natural language outputs are notoriously difficult to evaluate with popular metrics such as BLEU [40] and ROUGE [33] having well-known shortcomings [6]. Hence, we propose to evaluate language understanding as a multiple-choice question-answering task, thereby using the well-defined benchmark metric of overall question-answering accuracy.

The second failure mode for a long-term video task is that the proposed task happens to actually be a short-term one - only disguised as a long-term task. To measure the intrinsic "long-term" nature

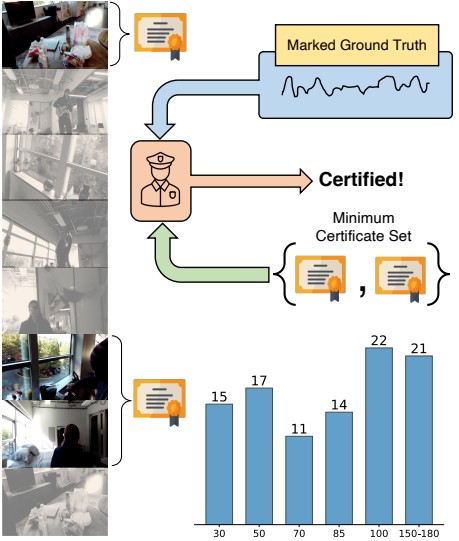

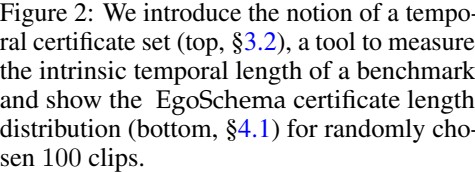

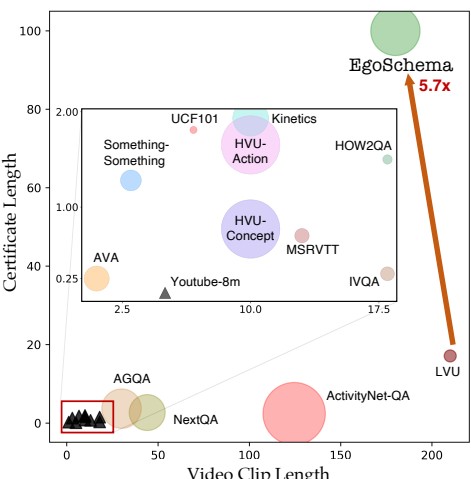

Figure 2: We introduce the notion of a temporal certificate set (top, §3.2), a tool to measure the intrinsic temporal length of a benchmark and show the EgoSchema certificate length distribution (bottom, §4.1) for randomly chosen 100 clips.

Figure 3: **Certificate Length across video datasets** for a broad spectrum of tasks such as action classification, detection, relationship classification, concept classification, video classification, and multiple choice question-answering. §4.1 details the precise operationalizations.

of a video understanding task, we propose the notion of temporal *certificate length* [4]. Intuitively, certificate length (§3.2) is the length of the video a human verifier needs to observe to be convinced of the veracity of the marked annotation. The idea of temporal certificates is not limited only to question-answering or vision-language tasks but is applicable to several video understanding tasks, including pure vision tasks such as action classification, detection, or even temporal action localization.

Based on the length of the temporal *certificate*, we propose the following temporal understanding taxonomy for video tasks: Datasets with certificate length in the order of 1 second are termed short video tasks. Next, we name datasets with certificate length in the order of 10 seconds as, long-form video tasks. Finally, datasets with certificate length in the order of 100 seconds are termed as, very long-form video tasks. Fig. 3 presents estimates of the certificate lengths for a variety of datasets plotted against the temporal length of the video clip. We observe that the temporal certificate length is quite weakly correlated with the length of the video clip. This is due to the intentional design choice in defining the certificate set, which decouples the task of searching or retrieving the relevant sub-clip from a bigger clip from the task of visually understanding the retrieved sub-clip. And in this manner, using temporal certificate length as a metric for measuring the intrinsic temporal hardness of a dataset, avoids the failure mode of formulating an implicitly short-term task disguised as a long-term one. Section 3.2 details precise operationalizations for estimating the temporal certificate sets.

In summary, our contributions are three-fold. *First*, we propose the notion of temporal certificates, a broadly applicable notion that measures the intrinsic temporal hardness of clips in a video understanding dataset. We estimate temporal certificate lengths for a broad variety of existing datasets and show that EgoSchema has a median temporal certificate of about 100 seconds, which is 5× longer than the dataset with the second longest certificate length [51], and 25× to 100× longer than all other existing video understanding datasets (with or without language). *Second*, building upon the notion of temporal certificates, we introduce EgoSchema, a diagnostic benchmark for assessing the very long-form video understanding capability of multimodal video-language systems. *Third*, we benchmark both state-of-the-art video-language systems and humans in Zero-shot settings on EgoSchema to find that even the most advanced current video-language understanding systems consisting of billion of parameters achieve very low accuracy in long-from multiple-choice question-answering (< 33%) while humans achieve about 76% accuracy in the unconstrained setting.

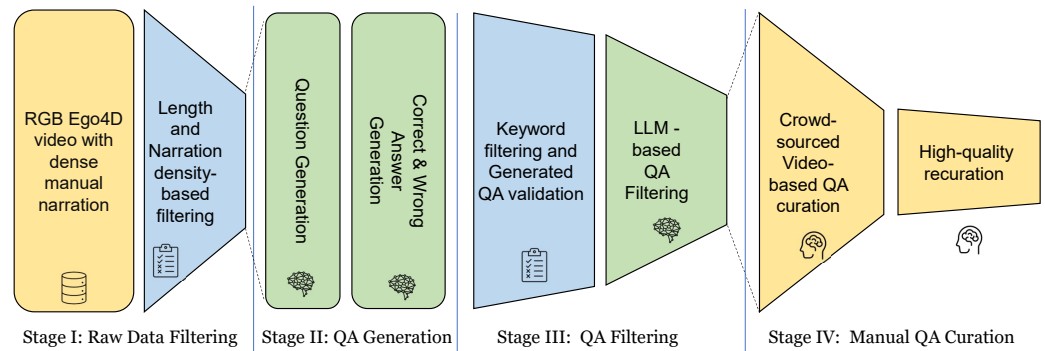

Figure 4: EgoSchema data pipeline. Stage I filters the suitable Ego4D RGB videos and narrations for question-answer generation (§3.1.1). Stage II uses narrations in a chained LLM prompting (§3.1.2) procedure to generate multiple $\mathcal{QAW}$ triplets per three-minute video clip (§3.1.2). Stage III performs pre-filtering with rule-based and LLM-based logic (§3.1.3). Finally, Stage IV involves two rounds of human curation on filtered $\mathcal{QAW}$ for selecting very long-form video-language understanding data (§3.1.4). The stage width ratios are indicative of the filter selection ratios.

## 2 Related Works

**Video Question-Answering Datasets.** Visual Question-Answering [3] is a popular video-language task with several large internet-scale datasets for video-language pre-training such as Ego4D [20], HowTo100M [35] and HowToVQA69M [34]. However, as the scope and size of pre-training datasets and models soar, it becomes critical to construct evaluations for assessing the model capabilities on various axes. Hence, many smaller datasets have been proposed for evaluating different aspects of video-language understanding such as compositional reasoning [21, 22], causal and common scene comprehension [53], instruction understanding [34, 57], video description ability [55], dynamic environments understanding [15], complex web video understanding [63], situated reasoning [50], spatiotemporal reasoning [28], social intelligence [65], dynamic neuro-symbolic reasoning [62], external knowledge-based reasoning [16] and many more [37, 61, 42, 10, 9, 13, 45, 56, 30, 31, 8, 64, 11, 32, 59, 66, 52, 24]. How2VQA69M [34] and iVQA [34] have leveraged HowTo100M [35] ASR text for generating questions. However, unlike Ego4D narrations that are used in EgoSchema, ASR text does not necessarily describe the visual elements in the scene. Hence, questions can suffer from biases where a key required information is visually absent. [25]. Additionally, generated question-answers also have quite short certificate lengths (iVQA in Fig. 2) due to the local nature of the ASR text.

**Long-form Video Understanding Datasets** have been very sparsely explored in prior works. [51] posits a long-form video understanding benchmark but the proposed tasks are unduly narrow and specific, such as the 'like' ratio and view count prediction. Also, [51] average certificate length is about $5.7\times$ smaller than EgoSchema.

[36] proposes a dataset for benchmarking efficient video inference consisting of frame-wise object mask annotations from Mask-RCNN [26] but without any long-term annotations. [43] introduces a dataset of about 111 hours of video sourced from Kinetics-400 [7] for generic event boundary detection. While the task itself requires comprehensive understanding, the video clip length is only 10 seconds long, with temporal *certificates* (§3.2) being much shorter. [47] proposes a question-answering dataset based on long movie clips but due to the open-ended nature of questions, successful approaches tend to neglect the visual data and are biased purely with approaches using additional text such as story lines. [44] proposes MAD, a language grounding dataset with an average clip of 110 minutes. However, the length of the retrieved clip is quite short (average 4.1 seconds) thereby resulting in a temporal *certificate* (§3.2) only a few seconds long. Further, MAD [44] and several other movie-based datasets [27, 48, 54] do not release any video data because of copyright issues. In contrast, EgoSchema has an average certificate length of about 100 seconds. Further, EgoSchema

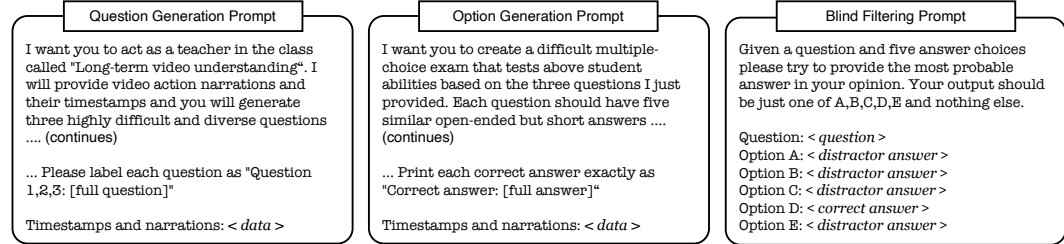

| Question Generation Prompt | Option Generation Prompt | Blind Filtering Prompt |
|---|---|---|
| I want you to act as a teacher in the class called "Long-term video understanding". I will provide video action narrations and their timestamps and you will generate three highly difficult and diverse questions .... (continues)

... Please label each question as "Question 1,2,3: [full question]"

Timestamps and narrations: < data > | I want you to create a difficult multiple-choice exam that tests above student abilities based on the three questions I just provided. Each question should have five similar open-ended but short answers .... (continues)

... Print each correct answer exactly as "Correct answer: [full answer]"

Timestamps and narrations: < data > | Given a question and five answer choices please try to provide the most probable answer in your opinion. Your output should be just one of A,B,C,D,E and nothing else.

Question: < question >
Option A: < distractor answer >
Option B: < distractor answer >
Option C: < distractor answer >
Option D: < correct answer >
Option E: < distractor answer > |

Figure 5: An abridged example of the generation and filtering prompts used in the EgoSchema data generation pipeline (§3). Full versions are provided in the *supplementary*.

will be publicly released under the Ego4D license, which allows direct public use of the video and text data for both research and commercial purposes.

# 3    Collecting EgoSchema

Collecting video and language datasets, even without a focus on very long-form video is quite challenging. Manually collecting, observing, and annotating videos with free-form language, in contrast to using images and pre-defined label categories, is both labor-intensive and time-consuming and thereby quite expensive. In addition to burgeoning cost, ensuring visual data diversity and minimizing visual and linguistic bias while ensuring high quality of marked annotations also contribute to the overall difficulty. All these factors get severely more challenging for long-form videos.

In this work, we propose a staged data collection pipeline (Fig. 4) utilizing existing large-scale but short-term video datasets, rule-based filtering procedures, and exciting new capabilities afforded by LLMs to significantly lighten the burden on human annotators. We use the proposed pipeline for curating EgoSchema, a high-quality and diverse very long-form video question-answering dataset. Associated datasheets [17] and data cards [41] for EgoSchema are provided in the *supplementary*.

## 3.1    EgoSchema Pipeline

### 3.1.1    Stage I: Raw Data Filtering

Ego4D [20] has over 3670 hours of RGB video spread consisting of over 3.85 million narration instances covering over 1,772 unique verbs (activities) and 4,336 unique nouns (objects) [20]. The narrators are instructed to continuously pause and describe everything that the camera wearer ('C') does. This creates dense and precise narrations that accurately describe the visuals.

Naturally, the collected video has non-uniform length and narration density. Since we would like to standardize the clip length for evaluation and have sufficiently rich narrations to allow interesting question-answer pairs to form in later stages, we filter the data based on the length and narration density. We choose to filter for non-overlapping three-minute clips each with at least 30 human annotated narrations (each narration is a timestamped sentence) to build EgoSchema. Detailed statistic of the number of viable clips for different possible length and narration density choices is discussed in *supplementary*.

### 3.1.2    Stage II: Question Answer Generation

The filtered narrations are processed with a capable LLM to generate $N$ Question-Answer triplets ($\mathcal{QAW}$), each consisting of the question $\mathcal{Q}$, the correct answer $\mathcal{A}$, and $M$ wrong answers $\mathcal{W}$, per clip. To achieve this, we experimented with several LLM inference call chaining procedures with trade-offs between quality and cost of generation that are briefly described next.

**One-shot** is the simplest prompting procedure to prompt for all $N$ instances of $\mathcal{QAW}$ in one inference call. This is the most cost-efficient option but we found the generations to be of significantly low quality. The generated $\mathcal{Q}$ often are very similar to each other and the generated $\mathcal{AW}$ have a very high false positive rate for the correct answers as well as a false negative rate for the wrong answers.

**N-shot** is the next natural prompting procedure where we generate one $\mathcal{QAW}$ per LLM inference call. This significantly improves the false positive and false negative rates but since the generated $\mathcal{Q}$ are independent and generated with the same prompt, they still tend to be very similar (comparable to one-shot), even at higher sampling temperatures. Further, the cost of generation also scales with $N$.

**QAW-shot** generates each of the $N$ questions $\mathcal{Q}$ in one inference call, followed by another inference call for generating $N$ correct answer $\mathcal{A}|\mathcal{Q}$ and finally, $N \times M$ wrong answers, $\mathcal{W}|\mathcal{Q}, \mathcal{A}$. Since each of the $N$ $\mathcal{Q}$ is generated jointly, they can be forced to be distinct with appropriate prompting. Similarly, the generated $\mathcal{A}$ and $\mathcal{W}$ can also be made distinct. However, this requires 3 *chained* LLM inference calls, and generation failures in earlier calls cascade steeply.

**Q(AW)-shot** generates each of the $N$ questions $\mathcal{Q}$ in one inference call, followed by a final inference call for generating all the $N$ correct and $N \times M$ incorrect answers in one go $\mathcal{A}, \mathcal{W}|\mathcal{Q}$. It enjoys the same uniqueness properties as QAW-shot while having just two chained calls, making it both $30\%$ cheaper and less prone to generation failure cascading. Further, between Q(AW)-shot and QAW-shot, we observe Q(AW)-shot to have a higher generated $\mathcal{A}$ quality, perhaps since LLM can jointly model $\mathcal{W}$ while generating $\mathcal{A}$. We choose this to be our main method of choice for generating $\mathcal{QAW}$.

**Prompt** for imputing narrations into the LLM has a tremendous effect on the quality of generated $\mathcal{QAW}$. We experiment with several seed prompts for each of which we inspect the quality of the $N$ generated $\mathcal{QAW}$ for 10 clips. Based on this we iteratively improve the seed prompts manually in a zeroth order optimization fashion. In total, we experiment with a total of about 85 prompts in this fashion to arrive at our final EgoSchema prompts – $\mathcal{P}_\mathcal{Q}$ for generating $N \times \mathcal{Q}$ questions and $\mathcal{P}_{\mathcal{AW}}$ for generating all remaining options $(\mathcal{AW})|\mathcal{Q}$. While we fix the $\mathcal{P}_\mathcal{Q}$ prompt, we use multiple $\mathcal{P}_{\mathcal{AW}}$ prompts so as to avoid any unintended bias in the options. Fig. 5 shows an abridged example of $\mathcal{P}_\mathcal{Q}$ and $\mathcal{P}_{\mathcal{AW}}$, full versions available in *supplementary* material.

**Choice of LLM** is extremely crucial for obtaining interesting long-form $\mathcal{Q}$ and generating hard negatives for $\mathcal{W}$. With weaker LLMs, the $\mathcal{Q}$ diversity across video clips remains narrow, and $\mathcal{W}$ tends to be either obviously wrong or, too similar to $\mathcal{A}$ and thus a false negative. While we experimented with both GPT-3 [5] and ChatGPT [38] but only found good quality generated $\mathcal{QAW}$ at a high enough rate with GPT-4 [39], Bard [18], and Claude [2]. For details please see *supplementary*.

We generate $N = 3$ questions per three-minute clip as well as $M = 4$ wrong answers to every question in addition to the correct answer. We observe that larger $N$ or $M$ tends to generate similar questions and wrong answers putting unnecessary pressure on Stages III and IV for filtering.

### 3.1.3 Stage III: Generated Question Answer Filtering

While Stage II produces several high-quality $\mathcal{QAW}$, even the best LLM generations are prone to output format aberrations, hallucinations, and sometimes plain false outputs. Further, despite specific pinpointed prompts (Fig. 5), LLMs can fail to comply. Since, we want to ensure EgoSchema to be extremely high-quality and accurate, we set up several filtering rounds to ensure the correctness and high difficulty of questions.

**Rule-based filtering.** Keywords from the prompts such as 'long-term', 'narrations', 'timestamp' etc. can sometimes bleed into the generated $\mathcal{QAW}$ which are then discarded. The output generations can also fail to parse according to a specified format and are also then discarded and the concerned $\mathcal{QAW}$ is regenerated.

**LLM-based filtering.** While rule-based filtering weeds out logic errors, we would like to further enrich $\mathcal{QAW}$ before employing human labor. For example, we aim to ensure EgoSchema requires grounded visual reasoning to solve, and hence questions should not be answerable *ungrounded*, without carefully observing the video. Hence, we develop a "blind" baseline.

**Blind filtering baseline** employs LLM to guess the correct answer based on the question, without having access to the video narrations conditioned on the shown filtering prompt (Fig. 5). All such ungrounded questions that can be answered blindly are filtered out. This also ensures that generated $\mathcal{W}$ are indeed relevant and plausible answers to $\mathcal{Q}$, since otherwise, the LLM would be able to guess $\mathcal{A}$ based only on the setting of $\mathcal{Q}$. Note that this is overly restrictive since it is possible that a question is guessed correctly through chance and is not necessarily ungrounded. However, we choose to optimize precision over recall since the amount of filtered $\mathcal{QAW}$ is still large enough.

**No-$\mathcal{Q}$ baseline.** We also experimented with a No-$\mathcal{Q}$ baseline, where the LLM is prompted to guess the correct answer using the narrations but without the question $\mathcal{Q}$. This ensures that the wrong answers are relevant and plausible to the video clip. However, we found this baseline to have near random accuracy ($\sim 20\%$), highlighting the efficacy of Stage II. Hence, we decided to not use this filter in the final pipeline. Additional details including the full prompt are in *supplementary*.

### 3.1.4 Stage IV: Manual $\mathcal{QAW}$ Curation

While LLM filtering ensures that the generated $\mathcal{QA}$ relates to the video content, it's also necessary to ensure the veracity and a long temporal certificate length for every generated $\mathcal{QAW}$. This is achieved through a two-step manual curation process.

In the first round of curation, annotators are tasked with three primary responsibilities: **(A)** First, they verify that $\mathcal{Q}$ is well-formed and $\mathcal{A}$ is indeed the correct answer to $\mathcal{Q}$. **(B)** Next, they confirm that all the $M$ distractors, $\mathcal{W}$, are indeed wrong answers to $\mathcal{Q}$. **(C)** Finally, they ensure that the temporal certificate length for answering $\mathcal{Q}$ is at least 30 seconds.

A $\mathcal{QAW}$ is discarded if any of these three conditions are not met. This reduces the number of admissible questions by a factor of about $4\times$ to $5\times$ within the first round itself. Next is a second round of re-curation, to reinforce the conditions and guarantee data of the highest quality. We find that more than $97\%$ of the questions that pass the first round also pass the second round, speaking to the efficacy of the curation process. A crucial aspect of ensuring that the question assesses very long-form video-language understanding capabilities is the notion of temporal certificate length (condition (C) above), which we describe next. The detailed procedures for onboarding and training the human annotators, as well as the instructions for the curation process are provided in the *supplementary*.

## 3.2 Temporal Certificates

We define the temporal *certificate* of a given video in a video understanding task to be the minimum set of *subclips* of the video that are both *necessary* and *sufficient* to convince a human verifier that the marked annotation for that data (such as timestamps in temporal activity localization, class label in activity recognition or, the correct option in multiple-choice question-answering) is indeed correct, without having to watch the rest of the clip outside of the certificate set (Fig. 2). Naturally, we define certificate length to be the sum of the temporal lengths of the sub-clips present in the certificate set.

**Meta-rules.** Datasets often have implicit rules that apply uniformly across the entire dataset. We call these conventions meta-rules and allow the human verifier to be well aware of them. For example, in temporal action localization datasets [29], an implicit assumption is that the action to be localized in a contiguous sub-clip and hence can be uniquely determined by the start and end timestamps. Since this rule is valid for all data, we consider it to be a meta-rule.

A comprehensive understanding of *meta*-rules of a dataset is necessary for accurate estimation of the certificate set, and hence the certificate length. Otherwise, a spuriously long certificate might be necessary to ensure the veracity of the marked annotations. For example, consider the task of action classification on Kinetics-400. A valid meta-rule to be made available to the human verifier in this case is the mutual exclusivity of action classes i.e., each data point can belong only to one of the 400 classes present in Kinetics-400. Without this understanding, given, say a 10-second clip of a human skiing, the certificate set needs to necessarily encompass the entire 10 seconds since otherwise the human verifier might not be convinced that all of the other 399 actions are not occurring in the clip. However, with the knowledge of the label exclusivity meta-rule, the certificate length will be drastically reduced to just a fraction of a second since just observing the action of skiing in a few frames is sufficient for the human verifier to out-rule all other action classes.

**Certificate Conventions**. For small certificate lengths, it is difficult for humans to estimate the exact sub-clip timestamps to be included in the certificate set. Hence, we choose to have a minimum length of $0.1$ second for a certificate. Further, in the case of two non-contiguous certificates, we collapse them into one if their closest ends are $< 5$ seconds apart. In cases where a fact needs to be verified at several places throughout the video, we let the annotator make a reasonable judgment for the length of the certificate to be included as long as it follows the above conditions.

# 4 Benchmarking EgoSchema

## 4.1 Evaluating Certificate Lengths

Fig. 3 presents certificate lengths for a spectrum of tasks spread across 15 different datasets such as, action classification (Kinetics [7], Something-Something [19], UCF101 [46], HVU-Action [12]), detection (AVA [23]), relationship classification (LVU [51]), concept classification (HVU-Concept [12]), video classification (Youtube-8M [1]), Question-Answering (NextQA [53], AGQA [22], NextQA [53], IVQA [34], MSRVTT [55], ActivityNet-QA [63], EgoSchema). For EgoSchema we benchmark the certificate length for 5 hours of video data ($100\mathcal{QAW}$) chosen randomly. For each other dataset, we ensure that (A) each annotated label class (if applicable) has at least 1 data sample evaluated and, (B) at least two hours of human effort is applied. Fig. 2 shows the histogram of estimated EgoSchema temporal certificate lengths for the 100 clips.

Fig. 3 plots the certificate length against the actual clip length. We observe that EgoSchema has temporal certificate length $5.7\times$ longer than the second longest certificate length dataset, and $10\times$ to $100\times$ longer than all other video understanding datasets.

## 4.2 Evaluating Multiple-choice Question Answering on EgoSchema

In Table 6, We benchmark several state-of-the-art video-language models, with the intention of adding more models in the future, in a Zero-shot question-answering setting on EgoSchema. We evaluate each model in at least two settings. First is the conventional inference setting, where the model is assessed based on the same number of frames it was trained with. And second is a less challenging setting, where the model is tested on the maximum number of frames possible to execute inference with, using an 80G A100, without exceeding the GPU memory capacity. In both settings, frames are sampled uniformly from the input video clip.

**FrozenBiLM** [58] adapts frozen multi-modal encoders trained on web-scale data for the task of question answering and achieves state-of-the-art zero-shot QA accuracy across 8 video question-answering datasets. We choose the How2QA FrozenBilM model under both 10 and 90 frames.

**VIOLET** [14] a masked token modeling-based video language transformer that performs competitively on a variety of video-language tasks. We evaluate four of the best VIOLET models that are finetuned on different tasks for both 5 and 75 frames and choose the model with the best overall accuracy. More details are in *supplementary*.

Figure 6: **Benchmarking Zero-shot QA on EgoSchema**

| Model | Release | Inference Params | Evaluation Setting | QA Acc |
|---|---|---|---|---|
| Choosing the correct $\mathcal{A}$ uniformly at random | | | | 20.0% |
| FrozenBiLM [58] | Oct 2022 | 1.2B | 10 frames | 26.4% |
| | | | 90 frames | 26.9% |
| VIOLET [14] | Sept 2022 | 198M | 5 frames | 19.9% |
| | | | 75 frames | 19.6% |
| mPLUG-Owl [60] | May 2023 | 7.2B | 1 frame | 27.0% |
| | | | 5 frames | 30.2% |
| | | | 10 frames | 29.7% |
| | | | 15 frames | 28.7% |
| | | | 30 frames | 20.0% |
| InternVideo [49] | Dec 2022 | 478M | 10 frames | 31.4% |
| | | | 30 frames | 31.8% |
| | | | 90 frames | **32**% |

**mPLUG-Owl** [60] proposes a training strategy to add image & video modality to pretrained large language models. We adapt mPLUG to facilitate the multiple choice QA by prompting the model with each of the options individually in the format: 'Given question `<question text>`, is answer `<answer text>` correct?' along with the video frames. Then, we choose the option with the highest softmax score of the token 'Yes' in the output text. We observe accuracy to be non-monotonic in frame length, and report results in 1 to 30 frames in Table 6.

**InternVideo** [49] proposes training video-language models jointly with masked video modeling and contrastive learning objectives. By default, InternVideo does not directly support multiple-choice video QA. We adapt the MSRVTT finetuned InternVideo model, which performs zero-shot

multiple-choice tasks, by incorporating the question with each answer choice in the format: 'Question: <question text>? Is it <answer text>'. Then, we choose the option with the highest output score as the prediction. We report results spanning 10 to 90 input frames in Table 6. We observe that performance is monotonic with the number of frames but the gain saturates around just 30 frames.

**Human.** We also benchmark human performance on multiple-choice question answering task on EgoSchema in Table 7. *First*, are time pressure settings where the annotators are asked to choose the correct answer under one ('In <1 min') and three ('In <3 min') minutes. Humans can already achieve an impressive 67.0% accuracy, in under 1 minute! Interestingly, this only slightly increases (+1.0%) when allowed three minutes. We believe that this can inform about performance on EgoSchema in limited model inference capacities. We believe this could inform about the frame rate needed for long-form video understanding in future models. *Second*, we also benchmark human performance using only 1 fps video ('180 frames'). Surprisingly, we observe that just with 1 fps humans can achieve an impressive 67.2%.

*Third*, we evaluate human performance in a restrictive setting where the annotator is forced to first watch the video without reading the text, and then answer the question without rewatching the video ('Video ⟶ Text'). Curiously, this achieves better accuracy than the 'No constraint' setting where the annotators are asked to simply answer without any constraints (76.2% vs. 75.0%). A possible hypothesis is that watching the video without text allows the annotator to focus more closely on the video, thereby benefiting performance than the setting where the attention is somewhat divided between the text and video. We believe this will help us understand the performance trade-offs in the early vs. late fusion of video and text modalities for long-form video-language models. All accuracies are estimated over 5 hours of video.

Figure 7: **Human Accuracy on EgoSchema**

| Evaluation Setting | QA Accuracy |
| --- | --- |
| 180 frames | 67.2% |
| In <1 min | 67.0% |
| In <3 min | 68.0% |
| No constraint | 75.1% |
| Video → Text | **76.2%** |

## 5 Conclusion

We present EgoSchema, a novel diagnostic benchmark designed for assessing very long-form video-language understanding capabilities of modern multimodal models. We also introduce the notion of a temporal *certificate* set, a probe that can be applied to a wide array of video tasks and benchmarks for understanding their intrinsic temporal lengths. We estimate temporal certificates of 15 varied datasets and demonstrate EgoSchema to exhibit temporal certificate length approximately $5.7\times$ longer than the next longest dataset and $25\times$ to $100\times$ longer than all other video understanding datasets. We also benchmark several state-of-the-art models on EgoSchema and find their Zero-shot question-answering accuracy to be less than $33\%$ while humans achieve 76%. We believe that EgoSchema will play a key role in the development and evaluation of future very long-form video-language models.

**Limitations.** EgoSchema RGB clips are sourced from Ego4D [20] and inherit Ego4D egocentric video biases. Further, the text is carefully curated for veracity, there are inevitable text data distribution biases that can occur in LLM-generated outputs due to biases present in web-scale LLM training data. Finally, human curation itself is far from perfect and while we perform two rounds of curation to minimize false positives, the collected EgoSchema is most likely to inevitably contain some small mislabelled or ill-formed question-answer sets. We plan to host a crowd-sourced errata board to minimize human curation error over time with the support of the open-source research community.

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

# EgoSchema Datasheet

**For what purpose was the dataset created?** Was there a specific task in mind? Was there a specific gap that needed to be filled? Please provide a description.

EgoSchema is a diagnostic benchmark for assessing very long-form video-language understanding capabilities of modern multimodal systems. While some prior works have proposed video datasets with long clip lengths, we posit that merely the length of the video clip does not truly capture the temporal difficulty of the video task that is being considered. To remedy this, we introduce temporal certificate sets, a general notion for capturing the intrinsic temporal understanding length associated with a broad range of video understanding tasks & datasets. Please see Section 3.2 in the main paper for more details.

**Who created this dataset (e.g., which team, research group) and on behalf of which entity (e.g., company, institution, organization)?**

The authors created the dataset within the Malik Group at Berkeley AI Research, UC Berkeley. The authors created it for the public at large without reference to any particular organization or institution.

---

**Composition**

**What do the instances that comprise the dataset represent (e.g., documents, photos, people, countries)?** Are there multiple types of instances (e.g., movies, users, and ratings; people and interactions between them; nodes and edges)? Please provide a description.

Each instance in the dataset represents a 3-minute video and text that contains a question and five answer options.

**How many instances are there in total (of each type, if appropriate)?**

EgoSchema has a total of 5063 instances each containing one video, one question, and five answer options. You can see further statistics on the whole data on our website egoschema.github.io.

**Does the dataset contain all possible instances or is it a sample (not necessarily random) of instances from a larger set?** If the dataset is a sample, then what is the larger set? Is the sample representative of the larger set (e.g., geographic coverage)? If so, please describe how this representativeness was validated/verified. If it is not representative of the larger set, please describe why not (e.g., to cover a more diverse range of instances, because instances were withheld or unavailable).

The video component of our dataset derives from the broader Ego4D dataset. For our research, we selectively extracted non-overlapping three-minute segments from the Ego4D video data, each segment consisting of a minimum of 30 human-annotated narrations (where each narration refers to a timestamped sentence). Detailed statistic of the number of viable clips for different possible length and narration density choices is discussed in Supplementary Section 6. The selected subset is very diverse in human behavior as can be seen by the activity statistics presented on egoschema.github.io.

**What data does each instance consist of? "Raw" data (e.g., unprocessed text or images) or features?** In either case, please provide a description.

Each instance in our dataset comprises raw mp4 video data, captured at a rate of 30 frames per second and with a high resolution. Accompanying this video data, there are six text elements - one question and five corresponding answer options one of which is marked as the correct answer to the question.

**Is there a label or target associated with each instance?** If so, please provide a description.

Each instance is associated with a label ranging from 1 to 5 that indicates which of the five answer options is correct.

**Is any information missing from individual instances?** If so, please provide a description, explaining why this information is missing (e.g. because it was unavailable). This does not include intentionally removed information but might include, e.g., redacted text.

All instances are complete.

**Are relationships between individual instances made explicit (e.g., users' movie ratings, social network links)?** If so, please describe how these relationships are made explicit.

Some instances may have the same video but different questions and answers. It will be indicated by a clip unique identifier in the final dataset.

**Are there recommended data splits (e.g., training, development/validation, testing)?** If so, please provide a description of these splits, explaining the rationale behind them.

EgoSchema is designed specifically for zero-shot testing. Its primary purpose is to be able to asses the out of the box long-term video-language understanding capabilities of modern multimodal models.

**Are there any errors, sources of noise, or redundancies in the dataset?** If so, please provide a description.

The dataset was very carefully manually curated to mitigate any incidence of errors within the questions and answers. Although different questions may be posed for the same clip, it is ensured that there is no overlap between any two distinct clips. Further related details are also discussed in the limitations section in the main paper.

**Is the dataset self-contained, or does it link to or otherwise rely on external resources (e.g., websites, tweets, other datasets)?** If it links to or relies on external resources, a) are there guarantees that they will exist, and remain constant, over time; b) are there official archival versions of the complete dataset (i.e., including the external resources as they existed at the time the dataset was created); c) are there any restrictions (e.g., licenses, fees) associated with any of the external resources that might apply to a future user? Please provide descriptions of all external resources and any restrictions associated with them, as well as links or other access points, as appropriate.

Entirety of the dataset will be made publicly available at our project website egoschema.github.io. We will also provide a download tool for preprocessing all the videos such as cutting clips, associating the question/answer text etc. Text will be released in a JSON format, hosted on our github repository. EgoSchema will be publicly released under the Ego4D license, which allows public use of the video and text data for both research and commercial purposes.

**Does the dataset contain data that might be considered confidential (e.g., data that is protected by legal privilege or by doctor-patient confidentiality, data that includes the content of individuals non-public communications)?** If so, please provide a description.

No

**Does the dataset contain data that, if viewed directly, might be offensive, insulting, threatening, or might otherwise cause anxiety?** If so, please describe why.

No

**Does the dataset relate to people?** If not, you may skip the remaining questions in this section.

Some videos do contain people. However, the Ego4D authors employed an array of de-identification procedures primarily centered on ensuring a controlled environment with informed consent from all participants, and, where applicable, in public spaces with faces and other personally identifiable

information suitably obscured. We strictly import all RGB information from Ego4D without any addition of our own.

**Does the dataset identify any subpopulations (e.g., by age, gender)?** If so, please describe how these subpopulations are identified and provide a description of their respective distributions within the dataset.

No

**Is it possible to identify individuals (i.e., one or more natural persons), either directly or indirectly (i.e., in combination with other data) from the dataset?** If so, please describe how.

No, Ego4D has employed an array of deidentification procedures in order to obscure any personally identifiable information such as people's faces.

**Does the dataset contain data that might be considered sensitive in any way (e.g., data that reveals racial or ethnic origins, sexual orientations, religious beliefs, political opinions or union memberships, or locations; financial or health data; biometric or genetic data; forms of government identification, such as social security numbers; criminal history)?** If so, please provide a description.

No

---

### Collection Process

**How was the data associated with each instance acquired?** Was the data directly observable (e.g., raw text, movie ratings), reported by subjects (e.g., survey responses), or indirectly inferred/derived from other data (e.g., part-of-speech tags, model-based guesses for age or language)? If data was reported by subjects or indirectly inferred/derived from other data, was the data validated/verified? If so, please describe how.

The video data, which is directly observable, was procured from the publicly accessible Ego4D dataset. In contrast, the text data was generated through the use of Large Language Models (LLMs) including GPT4, BARD, and Claude. These LLMs employed visual narrations from each video within the Ego4D dataset to generate the corresponding text.

**What mechanisms or procedures were used to collect the data (e.g., hardware apparatus or sensor, manual human curation, software program, software API)?** How were these mechanisms or procedures validated?

The video and narration data were downloaded in accordance with the official Ego4D guidelines for data access: https://ego4d-data.org/docs/start-here. For the generation of the text data within our dataset, we utilized API access for GPT4 via OpenAI, for BARD via Google, and for Claude via Anthropic. This allowed us to generate three distinct questions for each video clip sampled from the Ego4D dataset. Upon the generation of these questions for each sampled video clip, we implemented a series of filtering procedures including Rule-based filtering, Blind filtering, and Manual curation. See Section 3.1.2 in the main paper for a more detailed explanation.

**If the dataset is a sample from a larger set, what was the sampling strategy (e.g., deterministic, probabilistic with specific sampling probabilities)?**

The video component of our dataset derives from the broader Ego4D dataset. For our research, we selectively extracted non-overlapping three-minute segments from the Ego4D video data, each segment consisting of a minimum of 30 human-annotated narrations (where each narration refers to a timestamped sentence). Detailed statistic of the number of viable clips for different possible length and narration density choices is discussed in Supplementary Section 6.

**Who was involved in the data collection process (e.g., students, crowdworkers, contractors) and how were they compensated (e.g., how much were crowdworkers paid)?**

Our research utilized the services of Quantigo, a specialized data labelling company. The teams of Quantigo employees that were based in Bangladesh were compensated at a rate of 5 dollars per hour, at a wage significantly higher than the market hourly rate in Bangladesh. This was done to ensure fair compensation for the complex tasks performed while also contributing to the highest quality of the work delivered. It's important to note that our collaboration with Quantigo followed ethical guidelines, with the fair treatment of all employees involved and the appropriate respect for their expertise and labor. For exact instructions for human curation, see Supplementary Section 7.

**Over what timeframe was the data collected? Does this timeframe match the creation timeframe of the data associated with the instances (e.g., recent crawl of old news articles)?** If not, please describe the timeframe in which the data associated with the instances was created.

The original videos within the Ego4D dataset were collected across various occasions spanning from 2019 to 2021. As for the EgoSchema, the textual information was collected over several sprints during the first half of 2023 based on the Ego4D narrations.

**Were any ethical review processes conducted (e.g., by an institutional review board)?** If so, please provide a description of these review processes, including the outcomes, as well as a link or other access point to any supporting documentation.

No

**Does the dataset relate to people?** If not, you may skip the remaining questions in this section.

Yes

**Did you collect the data from the individuals in question directly, or obtain it via third parties or other sources (e.g., websites)?**

The video and narration data were acquired in accordance with the official Ego4D guidelines for data access: https://ego4d-data.org/docs/start-here/. The Ego4D authors had in turn ensured consent of the people involved.

**Were the individuals in question notified about the data collection?** If so, please describe (or show with screenshots or other information) how notice was provided, and provide a link or other access point to, or otherwise reproduce, the exact language of the notification itself.

Ego4d paper followed several procedures to ensure the preservation of privacy and the upholding of ethical standards. Notably, these procedures included obtaining informed consent from those wearing the cameras and adhering to de-identification requirements for personally identifiable information (PII). Given that the video collection was conducted by Ego4D, we are not in a position to provide specific instructions that were given to the camera wearers. The Ego4D privacy statement is available at https://ego4d-data.org/pdfs/Ego4D-Privacy-and-ethics-consortium-statement.pdf

**Did the individuals in question consent to the collection and use of their data?** If so, please describe (or show with screenshots or other information) how consent was requested and provided, and provide a link or other access point to, or otherwise reproduce, the exact language to which the individuals consented.

Ego4d paper privacy procedures have included obtaining informed consent from those wearing the cameras. Given that the video collection was conducted by Ego4D, we are not in a position to provide specific instructions that were given to the camera wearers. See Ego4D privacy statement.

**If consent was obtained, were the consenting individuals provided with a mechanism to revoke their consent in the future or for certain uses?** If so, please provide a description, as well as a link or other access point to the mechanism (if appropriate).

Ego4d paper privacy procedures have included allowing camera users to ask questions and withdraw at any time. Additionally, they were free to review and redact their own video. Given that the video collection was conducted by Ego4D, we are not in a position to provide specific instructions

that were given to the camera wearers. You can find the Ego4D privacy statement at https://ego4d-data.org/pdfs/Ego4D-Privacy-and-ethics-consortium-statement.pdf.

**Has an analysis of the potential impact of the dataset and its use on data subjects (e.g., a data protection impact analysis) been conducted?** If so, please provide a description of this analysis, including the outcomes, as well as a link or other access point to any supporting documentation.

While we recognize the importance of this topic, we would, once more, refer to the Ego4D paper for an in-depth discussion. Ego4D acknowledges the potential privacy risks associated with the use of wearable devices in data collection and has taken several steps such as depersonalizing any sensitive information, blurring out faces and bodies, etc. towards maintaining privacy. The same carries over to the video data in EgoSchema as well. We observed 50 randomly sampled three-minute clips from Ego4D, out of which we found 5 to have un-blurred and non-obstructed human faces (10%), collected with the participant's consent. Broadly, very long-form video understanding is a core capability for agents that are to perceive the natural visual world. Hence, developing datasets such as EgoSchema will be critical to unlocking this key AI capability. Additionally, according to Ego4D privacy statement, all videos from Ego4D were reviewed by an approved member of one of the participant's universities or institutes to identify and assess potential privacy concerns.

---

Preprocessing/cleaning/labeling

---

**Was any preprocessing/cleaning/labeling of the data done (e.g., discretization or bucketing, tokenization, part-of-speech tagging, SIFT feature extraction, removal of instances, processing of missing values)?** If so, please provide a description. If not, you may skip the remainder of the questions in this section.

The set of generated questions and answers from output was filtered by those LLMs and finally curated by humans. A detailed description can be found in Section 3. There was no preprocessing done on the video clips sampled from Ego4D.

**Was the "raw" data saved in addition to the preprocessed/cleaned/labeled data (e.g., to support unanticipated future uses)?** If so, please provide a link or other access point to the "raw" data.

Human curation was employed to rectify errors in the question-answer sets, particularly cases where the identified correct answer was wrong or a wrong answer was actually correct. Given the crucial role of this step in ensuring the accuracy of our dataset, we do not find it necessary to release a version of the dataset prior to human curation. However, all the discarded "raw" data is indeed also saved.

**Is the software used to preprocess/clean/label the instances available?** If so, please provide a link or other access point.

The APIs for the Large Language Models (LLMs) are publicly accessible. The prompts for filtering and instructions for human curation are provided in Supplementary Section 5 and Supplementary Section 7 respectively. Additionally all necessary code for generation, filtering etc. is provided in the supplementary materials.

---

Distribution

---

**Will the dataset be distributed to third parties outside of the entity (e.g., company, institution, organization) on behalf of which the dataset was created?** If so, please provide a description.

The dataset will be made publicly available and can be used for both research and commercial purposes under the Ego4D license.

**How will the dataset be distributed (e.g., tarball on website, API, GitHub)** Does the dataset have a digital object identifier (DOI)?

The dataset will be distributed as a JSON file describing the unique identifier for each clip, the associated question, the five answer options, the label, and additional clip information that facilitates the tracing of the clip back to the original Ego4D data, such as the Ego4D video identification of the clip's source video, among other details. In addition, download tools to acquire and pre-process the video RGB data will also be provided on our website.

**When will the dataset be distributed?**

The full dataset will be made available upon the acceptance of the paper before the camera-ready deadline.

**Will the dataset be distributed under a copyright or other intellectual property (IP) license, and/or under applicable terms of use (ToU)?** If so, please describe this license and/or ToU, and provide a link or other access point to, or otherwise reproduce, any relevant licensing terms or ToU, as well as any fees associated with these restrictions.

EgoSchema will be publicly released under the Ego4D license, which allows direct public use of the video and text data for both research and commercial purposes.

**Have any third parties imposed IP-based or other restrictions on the data associated with the instances?** If so, please describe these restrictions, and provide a link or other access point to, or otherwise reproduce, any relevant licensing terms, as well as any fees associated with these restrictions.

No

**Do any export controls or other regulatory restrictions apply to the dataset or to individual instances?** If so, please describe these restrictions, and provide a link or other access point to, or otherwise reproduce, any supporting documentation.

No

| Maintenance |
|:-----------:|

**Who will be supporting/hosting/maintaining the dataset?**

The authors of the paper will be maintaining the dataset, pointers to which will be hosted on github repo https://github.com/egoschema/EgoSchema along with the code for download and preprocessing tool, with the actual data hosted either on Amazon AWS as an S3 bucket or as a google drive folder.

**How can the owner/curator/manager of the dataset be contacted (e.g., email address)?**

We will post the contact information on our website. We will be available through github issues as well as through email.

**Is there an erratum?** If so, please provide a link or other access point.

We will host an erratum on the Github repo in the future, to host any approved errata suggested by the authors or the video research community.

**Will the dataset be updated (e.g., to correct labeling errors, add new instances, delete instances)?** If so, please describe how often, by whom, and how updates will be communicated to users (e.g., mailing list, GitHub)?

Yes, we plan to host an erratum publicly. There are no specific plans for a v2 version, but there does seem plenty oppurtunities for exciting future dataset work based on EgoSchema.

**If the dataset relates to people, are there applicable limits on the retention of the data associated with the instances (e.g., were individuals in question told that their data would be retained for a fixed period of time and then deleted)?** If so, please describe these limits and explain how they will be enforced.

No.

**Will older versions of the dataset continue to be supported/hosted/maintained?** If so, please describe how. If not, please describe how its obsolescence will be communicated to users.

N/A There are no older versions at the current moment. All updates regarding the current version will be communicated via our website.

**If others want to extend/augment/build on/contribute to the dataset, is there a mechanism for them to do so?** If so, please provide a description. Will these contributions be validated/verified? If so, please describe how. If not, why not? Is there a process for communicating/distributing these contributions to other users? If so, please provide a description.

Contributions will be made possible using standard open-source tools, submitted as pull requests to the relevant GitHub repository. Moreover, we will provide information on how to trace sampled clips back to their original source within the Ego4D dataset. This will enable users to access additional Ego4D data, such as narrations, summaries, and object detections, as applicable.

## Full Prompts

Here are some of the prompts we developed for generating EgoSchema.

### 5.1   Set A

#### 5.1.1   Question prompt

```
Input:
I want you to act as a teacher in the class called "Long-term video understanding".
    I will provide video action narrations and their timestamps and you will
    generate three highly difficult and diverse questions for your students about
    the high-level details in the video. You want to test students' following
    abilities:

Ability 1: Students' ability to summarize and compare long parts of the video
Ability 2: Students' ability to compress information from the video rather than just
     listing the actions that happened in the video.
Ability 3: Students' ability to identify the most important parts of the video.

Your questions should not mention any particular timestamps or narrations. Remember
    to make sure the correct answers to your questions do not list information from
     the narrations but compress them in a concise conclusion.

Examples of good and difficult questions:
"What is the main action of the video?"
"Why did C do action ...?"

AVOID the following types of questions:
"When ...?"
"How many ...?"
"How much ...?"

When announcing the question please label each question as "Question 1,2,3: [full
    question]"

Timestamps and narrations:
0 - C stares at the lamp
1 - C looks around the apartment
3 - C talks to a man
6 - C walks around the apartment
10 - C talks to a man
11 - C looks around the apartment
14 - a man plays the guitar
20 - a man walks around the apartment
21 - a man plays the guitar
21 - C walks around the apartment
22 - C stares at the window
24 - C looks around the apartment
27 - C talks to a man
29 - C picks chips from the table
32 - C looks around the apartment
36 - C stares at a man
36 - a man plays the guitar
37 - C walks around the apartment
41 - C sits on the sofa
43 - C stares at a man
44 - a man plays the guitar
49 - a man climbs up the sofa
50 - a man plays the guitar
62 - a man climbs down the sofa
63 - a man walks around the apartment
64 - a man plays the guitar
```

```
93 - C stares at the table
96 - a man plays a guitar
111 - C stares at the table
112 - a man walks around the apartment
114 - C stares at a man
115 - a man lifts the guitar
118 - a man walks around the apartment
120 - a man places guitar on the sofa
122 - a man walks around the apartment
125 - C stands up
126 - C walks around the apartment
131 - a man picks a coffee maker jug
132 - a man pours coffee in a cup
134 - C stares at a man
140 - C stares at the window
142 - C stares at a man
144 - a man drinks coffee
145 - C looks around the apartment
150 - C walks around the apartment
152 - C talks to a man
156 - C stares at a bench
158 - C looks around the apartment
164 - C stares at a man
164 - a man talks to C
167 - C looks around the apartment
171 - C follows a man
174 - a man points at the window
175 - C looks around the bedroom
================
Output:
```

Question 1: What can you deduce about the relationship between C and the man, based
on their interactions and behaviors throughout the video?

Question 2: Identify the primary activity in the apartment and discuss its
importance/significance. How does it influence the actions and atmosphere of
the video?

Question 3: Observe the dynamics of the video in terms of changes in characters'
actions, interactions, and spatial movement. How do these shifts contribute to
the overall narrative?

### 5.1.2 Answer prompt

```
Input:
```
I want you to act as a teacher in the class called "Long-term video understanding."
I will provide video action narrations and their timestamps and three highly
difficult and diverse questions for your students about the high-level details
in the video. I want you to test students' following abilities:

Ability 1: Students' ability to summarize and compare long parts of the video
Ability 2: Students' ability to compress information from the video rather than just
listing the actions that happened in the video.
Ability 3: Students' ability to identify the most important parts of the video.

I want you to create a difficult multiple-choice exam that tests above student
abilities based on the three questions I just provided. Each question should
have five similar open-ended but short answers, but only one should be correct.
Make it very difficult for students to find the correct answer among all the
wrong answers. All answers should be closely related to what happens in the
video. Make wrong answers significantly longer than correct answers. Ensure all
of the correct answers compress information from narrations them into a
concise conclusion. Your answers should not mention any particular timestamps
or narrations.

Do not use letters for the answer choices
Print each correct answer exactly as "Correct answer: [full answer]"
Please print each wrong answer on a new line and print each wrong answer as "Wrong
    answer 1,2,3,4: [full answer]"

Timestamps and narrations:
0 - C stares at the lamp
1 - C looks around the apartment
3 - C talks to a man
6 - C walks around the apartment
10 - C talks to a man
11 - C looks around the apartment
14 - a man plays the guitar
20 - a man walks around the apartment
21 - a man plays the guitar
21 - C walks around the apartment
22 - C stares at the window
24 - C looks around the apartment
27 - C talks to a man
29 - C picks chips from the table
32 - C looks around the apartment
36 - C stares at a man
36 - a man plays the guitar
37 - C walks around the apartment
41 - C sits on the sofa
43 - C stares at a man
44 - a man plays the guitar
49 - a man climbs up the sofa
50 - a man plays the guitar
62 - a man climbs down the sofa
63 - a man walks around the apartment
64 - a man plays the guitar
93 - C stares at the table
96 - a man plays a guitar
111 - C stares at the table
112 - a man walks around the apartment
114 - C stares at a man
115 - a man lifts the guitar
118 - a man walks around the apartment
120 - a man places guitar on the sofa
122 - a man walks around the apartment
125 - C stands up
126 - C walks around the apartment
131 - a man picks a coffee maker jug
132 - a man pours coffee in a cup
134 - C stares at a man
140 - C stares at the window
142 - C stares at a man
144 - a man drinks coffee
145 - C looks around the apartment
150 - C walks around the apartment
152 - C talks to a man
156 - C stares at a bench
158 - C looks around the apartment
164 - C stares at a man
164 - a man talks to C
167 - C looks around the apartment
171 - C follows a man
174 - a man points at the window
175 - C looks around the bedroom

Questions:

Question: what can you deduce about the relationship between c and the man, based on
    their interactions and behaviors throughout the video?

Question: identify the primary activity in the apartment and discuss its importance/
    significance. how does it influence the actions and atmosphere of the video?

Question: Observe the dynamics of the video in terms of changes in characters'
    actions, interactions, and spatial movement. how do these shifts contribute to
    the overall narrative?
================
Output:
Question 1: What can you deduce about the relationship between C and the man, based
    on their interactions and behaviors throughout the video?

Correct answer: They have a casual, friendly relationship.
Wrong answer 1: C and the man are complete strangers and never interacted before.
Wrong answer 2: C is deeply in love with the man, and the man reciprocates those
    feelings.
Wrong answer 3: They are business associates discussing a financial transaction.
Wrong answer 4: C is an intruder in the man's apartment, and the man is trying to
    get C to leave.

Question 2: Identify the primary activity in the apartment and discuss its
    importance/significance. How does it influence the actions and atmosphere of
    the video?

Correct answer: The primary activity is the man playing the guitar, which sets a
    relaxed environment.
Wrong answer 1: The main activity is C staring at different objects, indicating a
    deep curiosity.
Wrong answer 2: The primary activity is a heated argument between C and the man that
    escalates over time.
Wrong answer 3: The main activity is the man anxiously pacing around the apartment,
    creating a tense atmosphere.
Wrong answer 4: The primary activity is C and the man planning to commit a crime
    together.

Question 3: Observe the dynamics of the video in terms of changes in characters'
    actions, interactions, and spatial movement. How do these shifts contribute to
    the overall narrative?
Correct Answer: Actions and interactions are casual and relaxed, reflecting a
    comfortable environment.
Wrong Answer A : The video displays a profound sense of conflict and tension arising
    between the characters.
Wrong Answer B : The man is showing C the issues that need fixing in the apartment
    in a professional manner.
Wrong Answer C : Both the characters display an increasingly urgent need to solve an
    issue in the apartment.
Wrong Answer D : C and the man admire and interact with several objects in the
    apartment that look beautiful.

## 5.2   Set B

### 5.2.1   Question and answer prompt

Input:
I want you to act as a teacher in the class called "Long-term video understanding".
    I will provide video action narrations and their timestamps and you will
    generate three highly difficult and diverse questions for your students about
    the high-level details in the video. You want to test students' following
    abilities:

Ability 1: Students' ability to summarize and compare long parts of the video

Ability 2: Students' ability to compress information from the video rather than just
    listing the actions that happened in the video.
Ability 3: Students' ability to identify the most important parts of the video.

Your questions should not mention any particular timestamps or narrations. Remember
    to make sure the correct answers to your questions do not list information from
    the narrations but compress them in a concise conclusion.

Examples of good and difficult questions:
"What is the main action of the video?"
"Why did C do action ...?"

AVOID the following types of questions:
"When ...?"
"How many ...?"
"How much ...?"

When announcing the question please label each question as "Question 1,2,3: [full
    question]"

Timestamps and narrations:
3 - C holds the cloth in his right hand.
5 - the woman picks a carton from the grocery bag on the floor with her right hand.
6 - the woman drops the carton in a cabinet with her left hand.
7 - the woman dips both hands into the grocery bag.
9 - the woman drops a green carton on the floor with her right hand.
12 - C drops the green carton in the cabinet with his right hand.
13 - the woman holds a pack bag in her right hand.
16 - C opens a kitchen cabinet with his left hand.
18 - C removes a cereal pack from the kitchen cabinet with his left hand.
19 - C puts the green carton into the kitchen cabinet with his right hand.
21 - C closes the kitchen cabinet with his left hand.
24 - the woman removes a plastic from the grocery bag with her right hand.
25 - the woman drops the plastic on the floor with her right hand.
33 - C closes a wardrobe with his left hand.
38 - the woman puts a pack into the cabinet with her right hand.
43 - a dog lies down on a bed.
54 - C picks a cloth from the floor with his right hand.
58 - C adjusts the cloth with both hands.
66 - C hangs the cloth on the wall with both hands.
74 - the woman holds a grocery bag in her right hand.
82 - the woman touches her hair with her right hand.
92 - the woman talks with C.
99 - C holds two piece of cloths in both hands.
100 - the woman picks piece of clothes from a bag with both hands.
100 - C adjusts a camera on his head with his right hand.
103 - C drops the two piece of cloths on a couch with his left hand.
109 - C opens a door with his right hand.
110 - C walks into a toilet.
114 - C holds a red towel in his right hand.
116 - the woman talks with C.
118 - the woman scratches her back with her right hand.
118 - the dog walks into the toilet.
120 - C hangs the red towel on the wall with his right hand.
121 - C holds a towel in his left hand.
129 - the woman holds the towel in her left hand.
129 - the dog walks out of the toilet.
132 - C holds the door with his left hand.
133 - the woman touches her face with her right hand.
148 - C adjusts the camera on his head with his left hand.
149 - C touches his face with his right hand.
150 - the woman holds a bin with both hands.
160 - C opens the door with his right hand.
161 - C walks into the toilet.

```
163 - the woman walks into the toilet.
166 - the woman drops the bin on the floor with both hands.
168 - the woman adjusts the bin on the floor with both hands.
================
Output:
Question 1: Can you provide a concise summary of the primary activities taking place
    in the video, highlighting similarities between the actions of C and the woman?

Question 2: Based on the video, what task seems to be the central focus of both C
    and the woman's actions, and how do they contribute to the accomplishment of
    this task?

Question 3: Identify the three most important segments of the video and explain the
    significance of each in the overall context of the actions taking place.
================
Input:
Students are done with the questions. It is time to reveal the correct answers.
    Correctly provide short one-sentence answers to the questions you created, and
    additionally generate 4 very similar but wrong answers. Make sure that it will
    be hard to find the correct answer when combined with the wrong ones. Also,
    ensure that all the answers are concise and very close to an actual answer in
    length. Mainly make sure your correct answers do not list information from the
    narrations but compress them in a concise conclusion.

Do not use letters for the answer choices
Print each correct answer exactly as "Correct answer: [full answer]"
Please print each wrong answer on a new line and print each wrong answer as "Wrong
    answer 1,2,3,4: [full answer]"
================
Output:
Question 1:
Correct answer: C and the woman both interact with various objects to organize the
    space, including placing grocery items and hanging cloths.
Wrong answer 1: [Output disregarded]
Wrong answer 2: [Output disregarded]
Wrong answer 3: [Output disregarded]
Wrong answer 4: [Output disregarded]

Question 2:
Correct answer: The central focus is organizing and tidying up the space, with C and
    the woman putting items in their respective places.
Wrong answer 1: [Output disregarded]
Wrong answer 2: [Output disregarded]
Wrong answer 3: [Output disregarded]
Wrong answer 4: [Output disregarded]

Question 3:
Correct answer: Key segments include organizing groceries, adjusting household items
    (e.g., cloths, bins), and interacting with a dog.
Wrong answer 1: [Output disregarded]
Wrong answer 2: [Output disregarded]
Wrong answer 3: [Output disregarded]
Wrong answer 4: [Output disregarded]
```

### 5.2.2   Wrong answer prompt

```
Input:
I want you to act as a teacher in the class called "Long-term video understanding."
    I will provide video action narrations and their timestamps and three highly
    difficult and diverse questions for your students about the high-level details
    in the video. I will also show the correct answers to the questions. I want you
     to test students' following abilities:
```

Ability 1: Students' ability to summarize and compare long parts of the video
Ability 2: Students' ability to compress information from the video rather than just
    listing the actions that happened in the video.
Ability 3: Students' ability to identify the most important parts of the video.

I want you to create a difficult multiple-choice exam that tests above student
    abilities based on the three questions and their correct answers I just
    provided. Each question should have five similar open-ended but short answers,
    but only one should be correct. Make it very difficult for students to find the
    correct answer among all the wrong answers. All answers should be closely
    related to what happens in the video. Make wrong answers significantly longer
    than correct answers. Ensure all of the correct answers compress information
    from narrations them into a concise conclusion. Your answers should not mention
    any particular timestamps or narrations.

Do not use letters for the answer choices
Please print each wrong answer on a new line and print each wrong answer as "Wrong
    answer 1,2,3,4: [full answer]"

Timestamps and narrations:
3 - C holds the cloth in his right hand.
5 - the woman picks a carton from the grocery bag on the floor with her right hand.
6 - the woman drops the carton in a cabinet with her left hand.
7 - the woman dips both hands into the grocery bag.
9 - the woman drops a green carton on the floor with her right hand.
12 - C drops the green carton in the cabinet with his right hand.
13 - the woman holds a pack bag in her right hand.
16 - C opens a kitchen cabinet with his left hand.
18 - C removes a cereal pack from the kitchen cabinet with his left hand.
19 - C puts the green carton into the kitchen cabinet with his right hand.
21 - C closes the kitchen cabinet with his left hand.
24 - the woman removes a plastic from the grocery bag with her right hand.
25 - the woman drops the plastic on the floor with her right hand.
33 - C closes a wardrobe with his left hand.
38 - the woman puts a pack into the cabinet with her right hand.
43 - a dog lies down on a bed.
54 - C picks a cloth from the floor with his right hand.
58 - C adjusts the cloth with both hands.
66 - C hangs the cloth on the wall with both hands.
74 - the woman holds a grocery bag in her right hand.
82 - the woman touches her hair with her right hand.
92 - the woman talks with C.
99 - C holds two piece of cloths in both hands.
100 - the woman picks piece of clothes from a bag with both hands.
100 - C adjusts a camera on his head with his right hand.
103 - C drops the two piece of cloths on a couch with his left hand.
109 - C opens a door with his right hand.
110 - C walks into a toilet.
114 - C holds a red towel in his right hand.
116 - the woman talks with C.
118 - the woman scratches her back with her right hand.
118 - the dog walks into the toilet.
120 - C hangs the red towel on the wall with his right hand.
121 - C holds a towel in his left hand.
129 - the woman holds the towel in her left hand.
129 - the dog walks out of the toilet.
132 - C holds the door with his left hand.
133 - the woman touches her face with her right hand.
148 - C adjusts the camera on his head with his left hand.
149 - C touches his face with his right hand.
150 - the woman holds a bin with both hands.
160 - C opens the door with his right hand.
161 - C walks into the toilet.
163 - the woman walks into the toilet.

166 - the woman drops the bin on the floor with both hands.
168 - the woman adjusts the bin on the floor with both hands.

Questions:

Question: Can you provide a concise summary of the primary activities taking place
    in the video, highlighting similarities between the actions of c and the woman?
Correct answer: c and the woman both interact with various objects to organize the
    space, including placing grocery items and hanging cloths.

Question: Based on the video, what task seems to be the central focus of both c and
    the woman's actions, and how do they contribute to the accomplishment of this
    task?
Correct answer: the central focus is organizing and tidying up the space, with c and
     the woman putting items in their respective places.

Question: Identify the three most important segments of the video and explain the
    significance of each in the overall context of the actions taking place?
Correct answer: key segments include organizing groceries, adjusting household items
     (e.g., cloths, bins), and interacting with a dog.
================
Output:
Question 1: Can you provide a concise summary of the primary activities taking place
     in the video, highlighting similarities between the actions of c and the woman?

Correct answer: C and the woman both interact with various objects to organize the
    space, including placing grocery items and hanging cloths.
Wrong answer 1: C and the woman engage in lengthy conversations about their day
    while occasionally placing groceries in cabinets and adjusting cloths on walls.
Wrong answer 2: C and the woman focus on preparing an elaborate dinner, flipping
    through recipe books and chopping vegetables.
Wrong answer 3: Both C and the woman predominantly pay attention to the dog's well-
    being, providing it with food and water and playing with it.
Wrong answer 4: The video revolves around C and the woman working on a DIY project,
    with tools scattered around the room.

Question 2: Based on the video, what task seems to be the central focus of both c
    and the woman's actions, and how do they contribute to the accomplishment of
    this task?

Correct answer: The central focus is organizing and tidying up the space, with C and
     the woman putting items in their respective places.
Wrong answer 1: C and the woman are preoccupied with an ongoing debate, their
    actions aimlessly directed towards unrelated tasks.
Wrong answer 2: They are both primarily absorbed in their individual tasks,
    occasionally interacting with each other but with no clear common goal.
Wrong answer 3: The central focus of their actions is providing care and attention
    to the dog, and their actions revolve around its needs.
Wrong answer 4: The video primarily depicts a disagreement between C and the woman,
    with actions centered around this conflict.

Question 3: Identify the three most important segments of the video and explain the
    significance of each in the overall context of the actions taking place?

Correct answer: Key segments include organizing groceries, adjusting household items
     (e.g., cloths, bins), and interacting with a dog.
Wrong answer 1: The main segments involve discussing a movie, checking on an
    overheating stove, and attempting to catch a rodent in the room.
Wrong answer 2: The most important segments consist of C and the woman eating a meal
    , participating in a board game, and reading books.
Wrong answer 3: Three primary segments include C and the woman debating household
    chores, searching for lost items, and tending to a crying baby.
Wrong answer 4: Key segments include discussing an upcoming event, taking turns
    answering a phone call, and checking updates from an ongoing sports game.

# 6 Our clip length and narration density choice

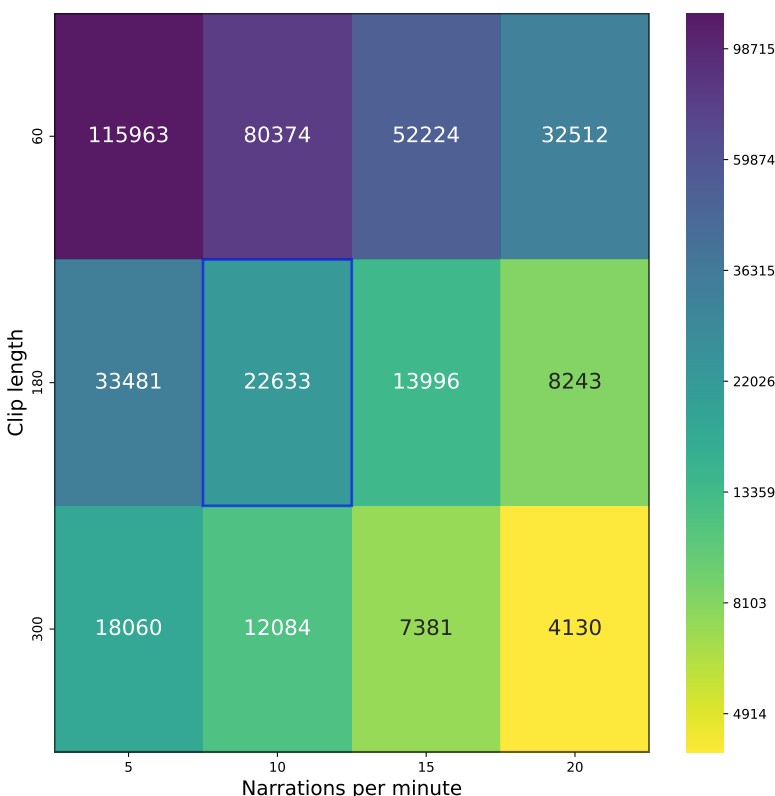

Figure 8: **Heatmap of a number of viable clips over a range of clip length and narration density.** There are only a few viable options that offer some degree of balance between the number of clips and the number of narration in the clip. One potential selection is to utilize 3-minute clips with a density of 5 narrations per minute, although this choice bears the significant disadvantage of potentially including clips with an insufficient volume of narration data to generate high-quality results. Another possible choice is to use 1-minute clips with a density of 20 narrations per minute, yet this option carries the drawback of the clips being too brief for the dataset to be very long-term. Hence, we choose 3-minute clips with a narration density of 10 narrations per minute as it offers a satisfactory balance between the number of narrations and clip length for generating EgoSchema.

# 7 Human curation

Our research utilized the services of a third-party company (not MTurk), for specifically training annotators to ensure quality. The process involved two distinct annotation procedures: data curation and human accuracy testing.

## 7.1 Curation

Generated data curation was performed by Quantigo employees. These curators were responsible for ensuring that the released EgoSchema dataset is tge highest high quality possible. Here is the exact instructions that was provided to annotators:

```
The annotation we need is to say that the Question-correct answer-wrong answer set (
    the whole set) is good if all these three conditions pass:

(Condition A) Question is Answerable: The question can be answered from the video
    and requires more than just a second of video to answer (so, if the answer is
    not present in the video or, if the answer can be formed with just a few frames
     (less than say, a second) then it fails this condition).
```

(Condition B) The Marked Correct Answer is correct: The ""correct answer"" is more
    the correct answer to the question

(Condition C) The Marked Wrong Answers are wrong: All 4 ""wrong answers"" are less
    correct than the ""correct answer"" (So for example, if a wrong answer is not
    completely false, but simply does not contain all the information that the ""
    correct answer"" does, then it is still a fine ""wrong answer"") IF even one of
    the marked answer is correct, the set should be labeled as bad.

(Condition D) The question is actually long-term: This is a very very important
    condition. We want the certificate for the question to be at least 30 seconds
    minimum. If the certificate is non-contiguous (ie. 5 seconds at one place, 20
    seconds at another, and 15 more seconds at a third place) the sum of lengths of
    all the sub-certificates together should be more than 30 seconds. Another
    example is, if a question can be answered simply from a few frames of the video,
    the certificate is small (and less than 30 seconds) and hence would fail this
    condition. Additional detials on how to handle certificate edge cases are
    provided in the annotator training through examples.

(Condition E) Avoid Boring Questions: Questions that ask about the frequency of
    something ("How many times..") fail this condition.

If any of these five conditions fail we want the whole set (Question / Correct
    Answer / Wrong Answer) marked bad.

Optional:
Since GOOD sets are so rare, in cases where it seems that a set is good but a small
    part of the above five conditions is not being met or, if one/two words were
    different this can be a good set, please label as MAYBE and we will fix it in
    the second round. We expect, Good/Bad to be about 97% of data and Maybe to be
    not more than 3%.

Extended notes:
1. In our experience, the wrong answers are made such that they differ from the
    correct answer in small but crucially meaningful ways. There are many cases
    where a careless reading of the wrong answer might make it seem that it is
    correct but upon careful inspection, it will become clear that something about
    the wrong answer indeed makes it wrong. While this is common, there are indeed
    cases where the wrong answer is also just as correct as the correct answer. In
    such cases, the wrong answer fails condition C, and the set is bad.

2. Roughly speaking, we expect about 20-25% of the questions that we have provided
    to be found as good. However, this is not necessary and the %age can be smaller
    or larger depending on each three-minute clip.

Edge Cases:
1. If the asked question has multiple answers and at least one of them aligns with
    the correct answer while none of them align with any of the other wrong answers,
    then provided that the top 5 conditions are met, we can mark the set as good.
2. If two questions are very similar (even within different clips) and both are GOOD
    , only choose one as GOOD and reject the other one with a comment mentioning
    this. We do not expect this to happen more than 1 or 2 times in a 100.
3. There might be more such edge cases, please feel free to contact me in such cases
    and we can expland.

# 8  Model evaluation by different temporal certificate length

We labeled temporal certificates for 86 clips and classified them into three distinct categories,
delineated by the length of the certificate: Category A (30 to 75 seconds), Category B (75 to 133
seconds), and Category C (133 to 180 seconds). These categories encompassed 37, 35, and 14
instances, respectively. Further, we evaluated the performance of 5 frame mPLUG-Owl, a state-of-
the-art 7 billion parameter model, across these defined categories. The accuracy rates were found
to be 35%, 20%, and 20% for Categories A, B, and C, respectively. These results underscore the

inherent complexities associated with longer temporal lengths in the context of model performance, even among advanced models. We hypothesize the reason for the accuracy drop in mPLUG-Owl to be the train test discrepancy the model faces in long videos in inference. mPLUG-Owl is trained with images only and uses per-frame image representations for representing video. Hence, with a large number of frames, the performance starts degrading since the network is not trained to handle it in training. We leave it to future work in long-term video understanding to further experimentally verify this hypothesis and mitigate this challenge.

**Certificate sampling.** Building upon the aforementioned experiment, we further extended our investigation by benchmarking the 5-frame mPLUG-Owl model specifically on the same corpus of 86 clips. However, in this analysis, frames were exclusively sampled from within the annotated temporal certificate. The derived accuracy for this method was 31.7%, which is 7% higher than the baseline accuracy of 24.7% that was obtained when frames were sampled uniformly throughout the entirety of the clip. This outcome further reinforces the significance and validity of the temporal certificate concept.

# 9    Annotator agreement.

To rigorously evaluate the robustness and consistency in the interpretation of the "temporal certificate" concept, we conducted an experiment involving two independent annotators. These annotators were tasked with evaluating 86 clips from EgoSchema, in order to measure the inter-annotator agreement using the intersection over union metric between the marked certificate sets. We find the IoU agreement to be 54.3% over the entire set. This denotes that while the exact certificate sets do have some subjectivity, annotators still largely agree on the long-term nature of certificates. Note that in the EgoSchema collection, we only use temporal certificates as a qualifying criterion (>30 seconds) and have a median length of 100 seconds.

# 10    Benchmarking details

## 10.1    Violet

Violet is a video language model comprised of a visual encoder, text encoder, and multimodal transformer pretrained on a variety of masked visual modeling tasks ranging from simple ones such as RBG pixel values up to more high levels ones such as spatially focussed image features. It performs competitively on a variety of video-language tasks such as Video-QA and Video-Text Retrieval. We evaluate one pre-trained model and 3 models finetuned on lsmdc-mc, msrvtt-qa, and msrvtt-retrieval. We evaluate using both 5 frames and 75 frames and choose the model with the best overall accuracy.

## 10.2    mPLUG-Owl

By default, mPLUG-Owl does not possess inherent capabilities for direct video question answering. As such, we undertook several experiments to adapt it to our required format. One approach involved inputting all answer choices in the form of a shuffled test. However, this resulted in a bias towards selecting the first option in most cases. For another approach, FrozenBiLM offered a methodology for frozen zero-shot models to operate in the context of multiple-choice video question answering, which inspired us to adapt this methodology for mPLUG-Owl. As mPLUG-Owl utilizes word-level tokenization, we could extract the confidence score for each generated token, particularly the 'Yes' token. We recorded the 'Yes' token confidence score for each answer option. In instances where the 'Yes' token was absent, we assigned the confidence score as zero, though empirically, in most cases, the model output was positive and contained the 'Yes' token. Ultimately, we selected the answer option with the highest 'Yes' confidence score as the model output given the question. In scenarios where multiple options scored the same highest confidence for the 'Yes' token, we randomly selected the answer from these top-scoring options. It should be noted that mPlug-Owl was originally trained to process a single image, and its capacity to handle additional frames is an emergent ability that has not been thoroughly tested to date."

## 10.3   InternVideo

The two most closely aligned formats supported by InternVideo are open-ended Video Question Answering and Zero-shot Multiple Choice tasks. In the case of open-ended Video Question Answering, the task is to predict the answer to a question posed within a video. However, due to the restricted vocabulary of open-ended answers in open-ended Video Questions Answering, we decided to formulate EgoSchema within the context of a Zero-shot Multiple Choice task. This task aims to identify the correct answer from a set of given options, without the inclusion of a question. InternVideo has provided finetuned weights for two datasets: MSRVTT and LSMDC. We selected the model finetuned on MSRVTT because it shares greater contextual similarity with EgoSchema.

## 10.4   Human

To conduct human benchmarking, we engaged a distinct team of ten employees within the same data annotation company to carry out human benchmarking on our dataset. The answers were randomized and presented in the form of a test. The following are the precise instructions provided to the annotators:

```
- Setting 1: Unlimited setting -- The goal is to get answers as accurately as
    possible without worrying about time.

- Setting 2: 1 minute timed setting -- In this case, the test taker (annotator) has
    only 1 minute to spend per question (including watching video/reading text/
    everything). If they do not have the answer, just guess the best based on their
     intuition and move on.

- Setting 3: 3 minutes timed setting-- Same as above but with 3 min instead of just
    one.

- Setting 4: Video -> Text setting -- In this case, the taker is not allowed to read
     the text before looking at the video and at the video after reading the text.
    In other words, the test taker can spend as much time as they want to look at
    the video first and then must move on to answering the question. They cannot go
     back to the video once they start reading the text. This is an untimed setting
     -- they can take as much time as they want per question.

- Setting 5: 180 frames setting -- This is the same as the untimed setting except
    the annotator has access to only 1 frame per second (ie the video feels like a
    GIF with one frame per second instead of the usual 30 frames per second) --
    each video is still 3 minutes long, but it feels more jittery. All instructions
     remain the same as in an untimed setting.
```

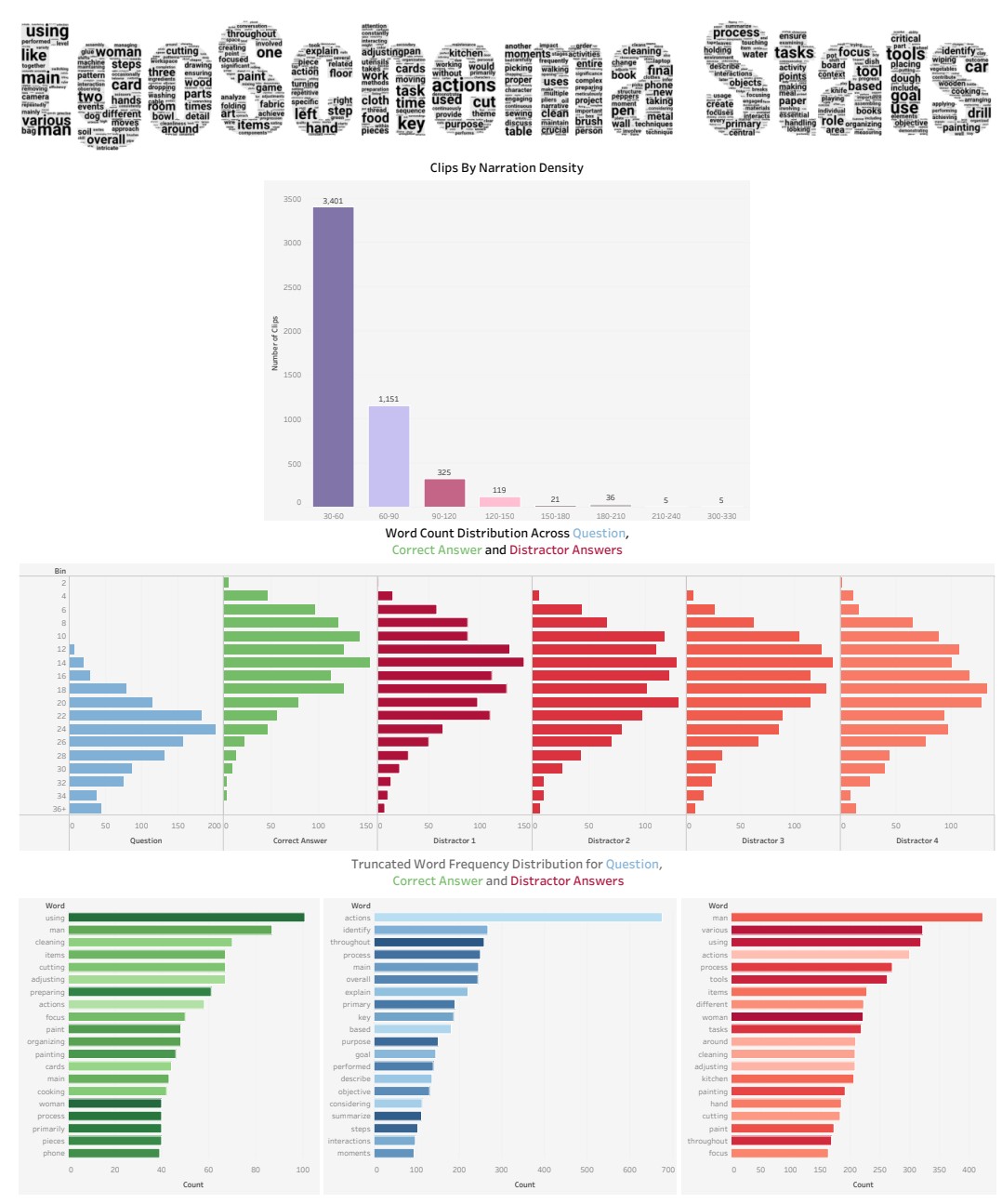

Figure 9: Interactive Version of these statistics visualizations can be found at the statistics page on our website.

