# OpenReview forum: "EgoSchema: A Diagnostic Benchmark for Very Long-form Video Language Understanding"
_NeurIPS.cc/2023/Track/Datasets_and_Benchmarks — NeurIPS 2023 Datasets and Benchmarks Spotlight_

### Official Review · Reviewer_uDzm · 2023-07-12

**Rating:** 7
**Confidence:** 2
**Clarity:** Yes

**Strengths:**

- Proposing to use temporal certificate sets to evaluate video understanding tasks provides an interesting perspective for future references.
- The combination of automated generation and human curation provides a scalable annotation approach.
- Open-sourcing of data and zero-shot model evaluation code under the Ego4D license at egoschema.github.io.
- Potential for EgoSchema to serve as a valuable evaluation probe for developing effective long-term video understanding systems in the future.


**Additional Feedback:**

NA

**Correctness:**

It appears to me that the dataset was constructed in a sound way and the evaluation methods and experiment design were appropriate and performed correctly.

**Documentation:**

The EgoSchema paper provides sufficient detail on data collection and organization, availability and maintenance, and ethical and responsible use.

**Ethics:**

I don't find any potential ethics concern here.

**Limitations:**

The authors of the EgoSchema paper have addressed the limitations of their work in Section 5 of the paper. They have not explicitly discussed the potential negative societal impacts of their work.

**Opportunities For Improvement:**

- Explore additional evaluation metrics beyond multiple choice question-answering
- Discuss the limitations of the dataset and models evaluated, especially the potential of domain generalization

**Relation To Prior Work:**

Yes, EgoSchema RGB clips are sourced from Ego4D dataset.

**Summary And Contributions:**

The EgoSchema paper presents a benchmark dataset for evaluating the long video understanding capabilities of modern vision and language systems. The dataset contains over 5000 human curated multiple choice question-answer pairs spanning 250 hours of real video data. The paper describes the creation of the dataset, including the use of temporal certificate sets to evaluate video understanding tasks. The paper also evaluates the performance of state-of-the-art video and language models on the dataset. Overall, the EgoSchema dataset is a comprehensive resource for understanding natural human activity and behavior.

---

> ### Author Response · Authors · 2023-08-20
> **Rebuttal**
>
> Thank you for your kind and thoughtful suggestions. We address the remarks below:
>
> **Other metrics**: While it is indeed possible to construct long-term Video QA as an open ended language generation task but we refrain from doing so since the metrics for measuring generation quality like BLEU score are quite noisy and imprecise (this is also noted in the paper introduction) The design choice of framing as a MC-QA problem allows using the clean accuracy score for evaluation. Another possible metric for evaluation is a log confidence based penalty score (similar to cross entropy) which would serve as a softer version of the hard 0-1 accuracy score and can indeed be explored as an interesting direction in future work on updating EgoSchema.
>
> **Limitations of the dataset and models evaluated, especially the potential of domain generalization**: Some of the limitations of the dataset are addressed in Section 5 of the paper. Also note that Ego4D is very diverse covering over 74 locations worldwide and 9 different countries. All this diversity both visual and textural is inherited into the domain of EgoSchema which is also quite diverse and has open-ended question-answering format. Could the reviewer please also clarify what aspect of domain generalization are they interested in for us to provide a more details answer?

---

### Official Review · Reviewer_vggB · 2023-07-20
**Review of EgoSchema Video Question Answering Dataset**

**Rating:** 8
**Confidence:** 5
**Clarity:** Yes, the paper is well-written and ea…

**Strengths:**

The dataset collection is carefully detailed within the paper and every step of the process is explained with reasoning behind why this is done. The step of generating the questions was nicely described between the different approaches and their drawback.

The introduced certificate length is a really nice concept on how to showcase differences in reasoning required for different datasets (and how they were annotated/tasks they were annotated for). This is another nice aspect to show the inherent difficulty of each dataset on a scale that has not been seen before.

**Additional Feedback:**

Line 301: closing quote mark is used instead of opening quote mark (`)

**Correctness:**

Yes, the dataset has been constructed in a sound way. Evaluation is appropriate and of sound design.

**Documentation:**

Yes, the datasheets for datasets have been provided within the supplementary material and a great deal of detail has been given within both the main paper and the supplementary material.

**Ethics:**

I don't believe there are any ethical concerns that warrant further discussion or review for this dataset.

**Limitations:**

Yes, the authors have discussed the potential limitations of the dataset including the bias from Ego4D as it is a labelled subset, the bias from using LLMs to generate questions, and the likelihood of a small number of errors even after human curation.

**Opportunities For Improvement:**

From looking at the Ego4D website, access to the videos (and the dataset) for Ego4D requires signing of their license. Is this also true for the EgoSchema benchmark?

Were videos chosen which all have blurred faces. The social benchmark of Ego4D looks like it does not have blurred face information but the datasheet says that "Ego4D has employed an array of deidentification procedures in order to obscure any personally identifiable information such as people’s faces". Were videos chosen for EgoSchema that exclude non-blurred faces?

[a] might be a nice reference for line 95 which investigated how alignable narrations are in HowTo100M and proves the issue that is discussed.

Are more results going to be added to Table 6 for the camera ready or is this in the future for wherever the dataset is hosted?

As the no-constraint results are over a different sized test set it might also be (part of) the reason there is a discrepancy between video->text and no-constraint, though I think the argument that forcing participants to watch the video first would provide some help. It would be best to report the human performance on the same 5 hours of video that the other settings use.



[a] Han, Tengda, Weidi Xie, and Andrew Zisserman. "Temporal alignment networks for long-term video." Proceedings of the IEEE/CVF Conference on Computer Vision and Pattern Recognition. 2022.

**Relation To Prior Work:**

The paper has clearly embedded itself within related work

**Summary And Contributions:**

In this paper a new dataset benchmark named EgoScheme is introduced. The benchmark is a video language understanding dataset that is designed for video-question answering to get over the issue of comparing generated sentences with the ground truth. The Ego4D dataset is used as a base with a LLM to generate questions and answers from the dense narrations. The curation process involves both an LLM and 2 rounds of human curation. Results show that state of the art methods are still a long way off human performance in the zero-shot setting (~30% compared to 76%).

---

> ### Author Response · Authors · 2023-08-20
> **Rebuttal**
>
> We thank the reviewer for their strong vote of confidence as well as their kind words in appreciating our work. We will add the provided reference [a] to the paper and address the concerns below:
>
> **License**: Yes, while we provide direct access to the videos on the release GitHub repo, we ask the users to kindly sign the permissive Ego4D license as well. This has been approved with the original licensees of Ego4D.
>
> **Blurred Faces**: Thanks for pointing the overlooked detail. Indeed we import faces of people that have provided their consent for recording faces to Ego4D into EgoSchema as well. On reviewer’s request we observe 50 randomly sampled three minute clips, out of which we find 5 to have un-blurred and non-obstructed human faces (10%), collected with the participant’s consent. We will add this detail to the datasheet.
>
> **New models**: Indeed, we will add more results for all the models released in the interim from paper submission deadline to CRM deadline in CRM. Further, we will maintain a public leaderboard with all the obtained results and allow future challenge participants to submit their results as well.
>
> **No-constraint results over the same 5 hours**: Thanks for the interesting observation. We do conduct the human experiments on the same five hours of the video data and find the accuracy to be 75.1% which is close to the original accuracy. We will add this detail to the paper as well.

---

> > ### Comment · Reviewer_vggB · 2023-08-30
> >
> > Thank you for providing your detailed reply to my review. After looking at the other reviews and responses I will be sticking with my original rating.

---

### Official Review · Reviewer_ZNHy · 2023-07-21

**Rating:** 8
**Confidence:** 4
**Clarity:** The paper is well written and easy to…

**Strengths:**

1. The authors propose evaluating temporal understanding difficulty based on certificate length rather than video clip length, providing valuable insights into the true challenge of long-form video tasks.
2. EgoSchema presents a challenging task with longer certificate length compared to previous video-language datasets, effectively assessing the capability of multimodal video-language systems in understanding very long-form videos.

**Additional Feedback:**

NA

**Correctness:**

The proposed dataset is constructed in a sound way, which leaverages both rule-based, LLM-based and human-based methods.

**Documentation:**

Yes.

**Ethics:**

No.

**Limitations:**

See above

**Opportunities For Improvement:**

1. It would be beneficial to analyze the certificate length of the predecessor dataset, Ego4D, to provide a comprehensive understanding of the benchmark's evolution.
2. More in-depth analysis of zero-shot QA on EgoSchema would enhance the paper, such as investigating why the QA accuracy drops as the number of frames increases from 5 to 30 in mPLUG-Owl, and exploring the reasons behind the marginal improvement in QA accuracy as the frame number increases.

**Relation To Prior Work:**

The paper clearly discussed how this work differs from previous contributions.

**Summary And Contributions:**

This paper present EgoSchema, a very long-form VideoQA benchmark. EgoSchema requires to select the correct answer from 5 options based on a 3-minute-long video clip. This benchmark is challenging since its intrinsic temporal understanding langth is very long (according to the proposed temporal certificate sets), and prevelant video-language models only achieve QA accuracy less than 33%.

---

> ### Author Response · Authors · 2023-08-20
> **Ego4D certificate and more 0-shot analysis**
>
> We thank the reviewer for kind words and recognition that temporal certificates “provide valuable insights into the true challenge of long-form video tasks.”
>
> **Certificate length of Ego4D**: Thank you for the great suggestion! We sampled clips from Ego4D natural language query task following identical certificate evaluation procedure as before and found the average clip length to be 495 seconds with an average certificate length of 9.6 seconds. We will also add this data point to the certificate plot.
>
>
> **More analysis of 0-shot acc**: Upon request, here we include two more analysis (that we will also add to the paper).
>
>
> - We benchmark the 0-shot accuracy by temporal certificate lengths. We label temporal certificates for 86 clips and group them into three categories based on certificate length: (A) 30 to 75 (B) 75 to 133 and (C) 133 to 180 seconds with 37, 35 and 14 questions belonging to each category respectively. We also benchmark mPLUG-OWL accuracies on the three formed categories where we find the accuracies to be 35%, 20% and 20% across the three categories respectively. This provides another evidence that longer temporal lengths are indeed challenging, including for SOTA 7B models.
> - We also benchmark 5 frame mPLUG-Owl on the above 86 clips but with only sampling frames from within the marked temporal certificate. This results is an accuracy of 31.7%, which is 7% higher than the baseline accuracy of 24.7% of sampling frames uniformly through the entire clip. Note that this is an oracle experiment since in test time the model do not have access to the temporal certificates and hence cannot sample from that set.
>
> We hypothesize the reason for accuracy drop in mPLUG-Owl to be the train test discrepancy the model faces in long videos in inference. mPLUG-Owl is trained with images only and uses per frame image representations for representing video. Hence, with a large number of frames the performance starts degrading since the text gets dominated with image tokens (which the network is not trained to handle in training). We leave it future work in long term video understanding to futher experimentally verify this hypothesis and mitigate this challenge.

---

> > ### Comment · Reviewer_ZNHy · 2023-08-21
> > **Response to the rebuttal**
> >
> > Thanks for the clarification. My problems are solved and I would like to further increase my score to 8: clear accept.

---

### Official Review · Reviewer_bgjf · 2023-07-21

**Rating:** 7
**Confidence:** 4
**Clarity:** The paper is easy to follow.

**Strengths:**

1. The introduction section is well-motivated as there is no suitable benchmark for multiple choice question answers regarding video temporal length, diversity of open-ended human behavior, and the intrinsic "long-term" nature. In addition, this paper is also easy to follow.
2. The idea of temporal certificates is reasonable since for long-form video QA, it is important to incorporate temporal information rather than a few frames. To validate the idea of temporal certificates, this paper calculates the certificate length across most current video datasets, including Kinetics400, ActivityNet, sth-sth, and other datasets for different tasks. The result for these current datasets is intuitive and the proposed Egochema is much longer than all these datasets.
3. The design of EgoSchema Pipeline is thoughtful since it not only includes a reasonable data filter, LLM-based QA generation and double-check, it also adopts manual curation-based detailed rules.
4. The benchmark is also challenging since these questions can not be answered only based on visual or textual features. This is due to the Blind-filter baseline and No-Q baseline check. I think it is useful to investigate the capability of visual-textual correspondence and reasoning for multimodal models.

**Additional Feedback:**

See "Opportunities For Improvement" and "Limitation".

**Correctness:**

The claims made in the submission are correct, and the dataset is constructed in a sound way.

**Documentation:**

Yes, there is there sufficient detail on data collection and organization, availability and maintenance, and ethical and responsible use. There is also sufficient detail to support reproducibility.

**Ethics:**

There are no ethical concerns with the submission that warrant further discussion or review.

**Limitations:**

This dataset is curated from public Ego4d datasets. and the Ego4d dataset has been processed to avoid potential negative societal impact by dropping related clips. In addition, this paper also plans to get help from the open-source research community for some small mislabeled ill-formed QA sets.

**Opportunities For Improvement:**

1. Ego4d is a long-form dataset of human daily activities, and the videos are continuous without shot transition. Therefore, some redundant descriptions like 'C walk around' and 'C turn right' in the datasets may interfere with models' understanding of actions and events. I think it may be better to filter these redundant texts for more meaningful cases.
2. In addition, I think the curated datasets should consider more about the distribution of action in original Ego4d datasets. In original datasets, most verbs are 'put', 'take', 'walk', etc. Without any related process, there may be a long-tail problem in the curated dataset.
3. There is no check for duplicate questions or answers generated by LLM for different clips. It is better to ensure the diversity of questions and answers when faced with different scenarios.
4. Some links in the paper are invalid.

**Relation To Prior Work:**

The paper discusses how EgoSchema differs from previous datasets in related works. Specifically, it has a much longer certificate length and adopts egocentric videos and language descriptions for a much more open-ended nature.

**Summary And Contributions:**

This paper curates a long-form video QA dataset from Ego4D, called EgoSchema, to assess multimodal models' very long-form video-language understanding abilities. Unlike other long-form video understanding datasets, EgoSchema also proposes new concepts, like temporal certificate length to indicate the length of the video for a human to observe to understand the accuracy of the marked annotation. It also analyses the differences between EgoSchema and other long-form datasets and quantitatively compares the differences using the proposed certificate length. This paper also benchmarks current SOTA video-language systems and humans in Zero-shot setting on EgoSchema to prove its difficulty.

---

> ### Author Response · Authors · 2023-08-20
>
> We appreciate the reviewer their detailed comments and deep understanding of our work. We will fix the broken link in CRM (because of a website update after the paper submission). Here are some follow on thoughts on the raised concerns:
>
> **Redundant descriptions**: Thanks for bringing up an interesting suggestion. Since we have two different sorts of models — the upstream LLM that generates questions and the downstream VLM that tackles long-term video QA let’s examine the effect of such description on each. For the LLM, since there is no direct access to the video we believe that such descriptions in fact serve as a crucial bridge for describing the video events to the LLM. And since the questions are generated away at a higher abstraction level by the LLM, such descriptions do not directly manifest in the questions. For the VLM, they do not interface with the captions themselves in EgoSchema but only the question and the answers and hence redundancy in captions do not directly affect their evaluations.
>
> **Long-tail problem**: This is perhaps both a bug and a feature. While long-tailed nature of distribution does hinder learning, the real world is, in fact, long tailed in several aspects. In fact for several crucial applications of AI, such as self-driving cars, the long tailed behavior of real world (such as events and human actions) is proving as one of the last remaining standing challenges. Since EgoSchema is a evaluation dataset only (and not meant for training) we believe that this long tail nature must be allowed to shine through in model evaluation for long-term video QA as well!
>
> That said, we do understand and empathize we the reviewer’s point of view. To that end, we must note that like for redundant descriptions, the atomic actions like ‘put’, ‘take’ etc are processed first with the LLMs (which are quite adept at long-tailed generalization) and are often abstracted away, thereby not directly manifesting in the output sentences.
>
> **Diversity of Questions**: We present a TSNE visualization of the question sentence embedding obtained from ada-002 model here: https://imgur.com/a/Y5eVtSh. The visualization shows the embedding for all the questions in the dataset. We observe that while questions do form loose clusters the questions still maintain decent diversity with only a few questions falling very close to each other.

---

> > ### Comment · Reviewer_bgjf · 2023-08-30
> > **Post-rebuttal**
> >
> > Thank you for your response and clarifications. I have understood your explanation for the effect of those redundant descriptions on the upstream LLM and the downstream VLM. I also agree that the VLMs should not be affected since they only interface with the questions and answers. However, it is better to provide some solid examples to better illustrate the effect for LLMs and how these descriptions serve as a crucial bridge for describing the video events.
> >
> > For the long-tailed problems, I agree with the fact that the real world is long-tailed in several aspects. But I think it is better to provide some ablation studies to investigate the effect of the long-tailed problem.
> >
> > Overall, the Egochema is a good benchmark to evaluate the longer-term reasoning abilities of video models, and I would like to keep my score at 7: Good paper, accept.

---

### Official Review · Reviewer_GM5p · 2023-07-21
**VideoQA benchmark for long-term video understanding**

**Rating:** 7
**Confidence:** 3

**Strengths:**

1. The underlying motivation to better assess and build a benchmark for longer-term video understanding capabilities of videoQA models is highly relevant to ongoing work in videoQA and has not been comprehensively explored before.

2. The introduced metric of 'temporal certificate length' is interesting and widely applicable beyond videoQA for general video-based reasoning tasks. Based on the metric, it seems that the proposed dataset requires much longer-term reasoning compared to existing video-based datasets.

3. The multi-step dataset curation process is well described and comprehensive to mitigate potential errors/biases and ensure high data quality. The usage of LLMs to form question-answer pairs is also novel, and the data and baselines evaluation is mentioned will be open sourced.

4. The zero-shot evaluation on the benchmark considers multiple recent models in various settings and also evaluates human performance in multiple settings, indicating that current videoQA models perform significantly lower than humans.

**Additional Feedback:**

N/A

**Clarity:**

Yes. Minor spelling/missing words can be checked again and fixed (e.g. L230 'action to be localized in' should be 'action to be localized is in'?).

**Correctness:**

Yes, the dataset curation and model evaluation is adequately described and appears correct.

**Documentation:**

Yes. Dataset and model evaluation code is stated will be released publicly.

**Limitations:**

Limitations are mentioned at the end.

**Opportunities For Improvement:**

1. It is not clear how the temporal certificate lengths are calculated both from writing and fig. 2. Do human annotators have to decide what are the salient clips (proposed minimum certificate set), which is then further verified by another human verifier (as seems to be indicated in fig. 2). Also, since this task can be quite subjective and is fairly complex, was inter-annotation agreement or an independent verification/quality check round utilized?

2. The current evaluated models are only tested in a zero-shot setting. It would be interesting to see how models would fair in a one-, five- or ten-shot setting for a more comprehensive evaluation of models and to also better assess the difficulty posed by the benchmark.

3. Further, it could also be interesting to see how models perform for different certificate lengths (perhaps for the 100 randomly chosen clips reported in fig. 2) .

**Relation To Prior Work:**

Yes, relevant works are referenced and differentiated.

**Summary And Contributions:**

This work proposes a video question answering benchmark derived from Ego4D specifically to evaluate the longer-term reasoning abilities of videoQA models. The task is framed as: given a question and a 3 minute video clip, choose the correct answer amongst 5 choices. The authors also introduce a metric termed 'temporal certificate set' to better assess the true temporal length (or the minimum set of salient video clips) a model or viewer will have to necessarily watch in order to be confident that a provided annotation is correct. Based on this metric, they report that while existing benchmarks/datasets may have lengthy video clips, they do not necessarily (and often don't) require reasoning over large temporal lengths for correct predictions. In contrast, the proposed EgoSchema is specifically designed such that longer temporal spans have to be considered for a model to be able to make a correct prediction, and based on the metric is shown to be so. The benchmark is intended as a zero-shot testing/probe set only (with over 5000 questions for more than 250hrs of real world videos) with no training set.

The dataset creation mechanism leverages on LLMs as well as humans/manual intervention in a multi-step manner to ensure the questions and associated multiple-choice answers are of high quality and with minimal biases. The evaluation of existing methods is also extensive (although currently done only in a zero-shot manner) and shows limitations of existing videoQA models in long-form video understanding.

---

> ### Author Response · Authors · 2023-08-20
> **Rebuttal**
>
> We thank the reviewer for their encouraging words and valuable comments and address the concerns below:
>
> **Consistency in Temporal Certificate Sets**: Upon suggestion from the reviewer we perform the following human experiment: We annotate a set of 86 clips with two annotators and measure the inter-annotator agreement using the intersection over union metric between the marked certificate sets. We find the IoU agreement to be 54.3% over the entire set. This denotes that while the exact certificate sets do have some subjectivity, annotators still largely agree on the long-term nature of certificates. Note that in the EgoSchema collection we only use temporal certificates as a qualifying criterion (>30 seconds) and have a median length ~100 seconds.
>
> **Few shot setting for Video QA**: While it is definitely an interesting direction to explore, it is worth noting that naive few shot learning with simple prompting (as done in GPT models in language) is quite non trivial with long video QA because of memory and compute bottlenecks. Even in a single video, the models are limited to far fewer frames than maximum available due to the same reason. Further, even if inference were to be done with more frames than used in training (as done in the mPLUG-OWL experiments in the paper) the performance becomes non monotonic with number of frames due to the train test distribution shift caused by using larger number of frames. Exploring efficient ways of performing few shot learning on EgoSchema is a promising direction for future research. To enable this, we will be releasing a small separate few shot prompting set (100 clips ~ 5 hours of video content) and maintain a separate leader-board for 1-shot, 2-shot and n-shot long-term video QA learning. Thanks for the great suggestion!
>
> **Performance by temporal certificate length**: We label temporal certificates for 86 clips and group them into three categories based on certificate length: (A) 30 to 75 (B) 75 to 133 and (C) 133 to 180 seconds with 37, 35 and 14 questions belonging to each category respectively. We also benchmark mPLUG-OWL accuracy on the three formed categories where we find the accuracy to be 35%, 20% and 20% across the three categories respectively. This provides another evidence that longer temporal lengths are indeed challenging, including for SOTA 7B models.

---

> > ### Comment · Reviewer_GM5p · 2023-08-28
> >
> > Thank you for the response and clarifications. I believe mentioning the inter-annotation scores and relative subjectivity of certificate lengths could be informative to readers. Further mentioning the few-shot benchmarking and specifying performance by temporal certificate length could better highlight current challenges in longer-term video understanding.
> >
> > I have raised my score to 7 as my concerns have been addressed.

---

### Decision · Program_Chairs · 2023-09-22

**Decision:**

Accept (Spotlight)

**Comment:**

The EgoSchema paper introduces a challenging long-form VideoQA benchmark using the Ego4D dataset. It proposes the concept of "temporal certificate length" to assess the required temporal understanding for correct predictions. The paper receives positive feedback on its motivation, dataset creation, and evaluation of existing models. However, some suggestions and opportunities for improvement are provided, such as filtering redundant descriptions in the dataset, analyzing the certificate length evolution, exploring more evaluation metrics, and addressing potential negative societal impacts. Additionally, clarity on the calculation of temporal certificate lengths and the usage of blurred-face videos is requested. The chair acknowledges the paper's strengths, including the comprehensive dataset creation process and the introduction of the novel temporal certificate concept.